# A live measles-vectored COVID-19 vaccine induces strong immunity and protection from SARS-CoV-2 challenge in mice and hamsters

Phanramphoei N. Frantz[1,2], Aleksandr Barinov [1,3], Claude Ruffié[1], Chantal Combredet[1], Valérie Najburg[1], Guilherme Dias de Melo [4], Florence Larrous [4], Lauriane Kergoat [4], Samaporn Teeravechyan[2], Anan Jongkaewwattana[2], Emmanuelle Billon-Denis[5], Jean-Nicolas Tournier [5], Matthieu Prot[6], Laurine Levillayer[6], Laurine Conquet[7], Xavier Montagutelli [7], Magali Tichit[8], David Hardy[8], Priyanka Fernandes[9], Hélène Strick-Marchand[9], James Di Santo [9], Etienne Simon-Lorière [6], Hervé Bourhy [4] & Frédéric Tangy [1✉]

Several COVID-19 vaccines have now been deployed to tackle the SARS-CoV-2 pandemic, most of them based on messenger RNA or adenovirus vectors.The duration of protection afforded by these vaccines is unknown, as well as their capacity to protect from emerging new variants. To provide sufficient coverage for the world population, additional strategies need to be tested. The live pediatric measles vaccine (MV) is an attractive approach, given its extensive safety and efficacy history, along with its established large-scale manufacturing capacity. We develop an MV-based SARS-CoV-2 vaccine expressing the prefusion-stabilized, membrane-anchored full-length S antigen, which proves to be efficient at eliciting strong Th1-dominant T-cell responses and high neutralizing antibody titers. In both mouse and golden Syrian hamster models, these responses protect the animals from intranasal infectious challenge. Additionally, the elicited antibodies efficiently neutralize in vitro the three currently circulating variants of SARS-CoV-2.

[1] Institut Pasteur, Université de Paris, Innovation Lab: Vaccines, Paris, France. [2] National Center for Genetic Engineering and Biotechnology (BIOTEC), National Science and Technology Development Agency, Virology and Cell Technology Laboratory, Pathumthani, Thailand. [3] Viroxis, Paris, France. [4] Institut Pasteur, Université de Paris, Lyssavirus Epidemiology and Neuropathology Unit, Paris, France. [5] Armed Forces Biomedical Research Institute (IRBA), Microbiology and infectious diseases department, Brétgny-sur-Orge, France. [6] Institut Pasteur, Université de Paris, Evolutionary Genomics of RNA viruses unit, Paris, France. [7] Institut Pasteur, Université de Paris, Laboratory of Mouse Genetics, Paris, France. [8] Institut Pasteur, Université de Paris, Experimental Neuropathology unit, Paris, France. [9] Institut Pasteur, Université de Paris, INSERM U1223, Innate Immunity unit, Paris, France. ✉email: frederic.tangy@pasteur.fr

The current pandemic of severe acute respiratory syndrome coronavirus 2 (SARS-CoV-2) responsible for COVID-19 pneumonia has already infected more than 200 million and killed over 4.7 million people worldwide at todays' date. The pandemic has resulted in unprecedented global social and economic disruption, with a projected "optimistic loss" of $3.3 trillion and a worst-case scenario loss of $82 trillion worldwide. While the virus is still expanding in a third wave in Europe, Latin America, and Asia, no specific treatment has been shown to prevent or cure the disease. Global development of effective vaccines that can prevent infection, disease, and transmission, together with enforcing public health protection measures, are the only ways to return to pre-COVID-19 normalcy. After exceptional commitment from the scientific and industrial communities over the past year, several vaccines have been successfully developed in an amazingly accelerated time frame, approved by health authorities, and effectively deployed in many countries. These vaccines based on messenger RNA (mRNA) and adenovirus vectors have demonstrated high levels of protection from COVID-19[1–3]. However, the duration of protection afforded by these vaccines is unknown, as well as their capacity to help controlling new emerging SARS-CoV-2 variants. Moreover, cold chain logistics or manufacturing issues complicate their global accessibility. A vaccine for the younger population, adolescents and children who are active transmitters of the virus, is still missing[4]. For these reasons, exploring other vaccine strategies is needed.

Numerous vaccine platforms are being used to develop SARS-CoV-2 vaccines[5]. Among them, live attenuated viral vectors look particularly interesting as they induce lasting protective immunity after a single dose and are inexpensive to manufacture at large scale. In particular, the live attenuated measles vaccine (MV) is one of the safest and most efficacious human preventive medicines. It elicits neutralizing antibodies and robust, long-lasting Th1 cellular responses, making it an attractive candidate for SARS-CoV-2 vaccination with minimal risk of vaccine-associated enhanced respiratory disease (VAERD)[6].

SARS-CoV-2 is an enveloped single-stranded positive-sense RNA virus belonging to the *Coronavidae* family and the *Betacoronavirus* genus[7]. Whole-genome sequencing of SARS-CoV-2 revealed 79.6% nucleotide sequence similarity with SARS-CoV-1[8]. The genome of SARS-CoV-2 encodes 4 structural proteins: the spike protein (S), the envelope protein (E), the membrane protein (M), and the nucleocapsid (N). The S protein, a trimeric class I fusion protein localized on the surface of the virion, plays a central role in viral attachment and entry into host cells. Cleavage of the S protein into S1 and S2 subunits by host proteases[9] is essential for viral infection. The S1 subunit contains the receptor-binding-domain (RBD), which enables the virus to bind to its entry receptor, the angiotensin-converting enzyme 2 (ACE2)[7,10]. After docking with the receptor, the S1 subunit is released and the S2 subunit reveals its fusion peptide to mediate membrane fusion and viral entry[11]. The coronavirus S protein contains the major epitopes targeted by neutralizing antibodies and is thus considered as a main antigen for developing vaccines against human coronaviruses[11–13]. Antibodies targeting the RBD may neutralize the virus by blocking viral binding to receptors on host cells and preventing entry. In addition, it has been observed that synthetic peptides mimicking and antibodies targeting the second heptad region (HR2) in the S2 subunit of SARS-CoV have strong neutralizing activity[14–16], likely by preventing the conformational changes required for membrane fusion. Efforts to develop a SARS-CoV-2 vaccine have thus focused on eliciting responses against the S protein.

A number of recombinant MV (rMV)-based vaccines against viral pathogens are currently in preclinical and clinical trials[17].

An rMV-based vaccine against chikungunya virus was demonstrated to be well-tolerated and immunogenic in phase I and II clinical trials, eliciting 90% seroconversion after a single immunization and 100% after boost despite the presence of preexisting measles immunity in volunteers[18,19]. Other MV-based candidates currently in clinical development include vaccines against Zika and Lassa viruses[20,21]. We also previously showed that rMV expressing the unmodified SARS-CoV-1 S protein induced a Th1-oriented response with high titers of neutralizing antibodies that protected immunized mice from infectious intranasal challenge by SARS-CoV-1[12]. An MV-MERS-CoV vaccine has also yielded promising preclinical results[22]. Given the excellent safety and efficacy profiles of these vaccine candidates, an MV-based vaccine targeting the S protein of SARS-CoV-2 has great potential to be both safe and effective. To explore this potential, we generated a series of rMVs expressing either full-length S or the S2 subunit protein of SARS-CoV-2 in prefusion-stabilized or native forms and tested their capacity to elicit neutralizing antibodies and T-cell responses in a mouse model of measles vaccination, and to protect immunized mice from intranasal challenge with mouse-adapted SARS-CoV-2. In addition, we tested the immunogenicity and protective efficacy of our lead candidate in the relevant golden Syrian hamster model of SARS-CoV-2 challenge[23].

## Results

**Design of SARS-CoV-2 S antigens.** Based on our previous work with MV expressing SARS-CoV-1 S, in which the surface-expressed full-length antigen showed higher immunogenicity[12] and since SARS-CoV and SARS-CoV-2 S proteins share a high degree of similarity[24], the full-length S protein of SARS-CoV-2 with transmembrane domain was chosen as the main antigen to be expressed by the MV vector. To improve its expression, we introduced a number of modifications in the native S sequence (Fig. 1), including human codon-optimization and mutation of two prolines, K986P and V987P, in the S2 region, following a proven strategy to stabilize the S protein in its prefusion conformation, increasing its expression and immunogenicity[25–27]. In addition, to increase the surface expression of the S protein in MV-infected cells, we deleted the 11 C-terminal amino acids (aa 1263-1273) from the S cytoplasmic tail (CT) to generate dER constructs.

To investigate the possibility of generating a broad-spectrum vaccine targeting SARS-CoV-2 and its emerging variants, we also designed antigens based on the S2 subunit (Fig. 1b) as this region is highly conserved among SARS-like CoVs and shares 99% identity with those of bat SARS-like CoVs (SL-CoV, ZXC21 and ZC45) and of a human SARS-CoV-1[24]. The antigens were generated both in its native trimer and prefusion-stabilized form, with the signal peptide of the S protein inserted in the N-terminus for the cell surface targeting. We designed a total of four different SARS-CoV-2 S constructs (Fig. 1b): (1) a native-conformation full-length S trimer (SF-dER); (2) a prefusion-stabilized full-length S (SF-2P-dER); (3) a native conformation S2 trimer (S2-dER); and (4) a prefusion-stabilized S2 (S2-2P-dER).

**Expression profile of SARS-CoV-2 S antigens.** Full-length S and S2 sequences were first cloned into the pcDNA mammalian expression vector and transfected into HEK293T cells to verify expression and assess surface protein localization by surface staining followed by flow cytometry. Prefusion-stabilized S constructs were observed to localize more strongly to the surface of transfected cells (Supplementary Fig. 1). Functionality of the S proteins was analyzed by transfecting the same pcDNA vectors in Vero cells, which express ACE-2. Activation of the fusion protein can be observed through the formation of large syncytia among

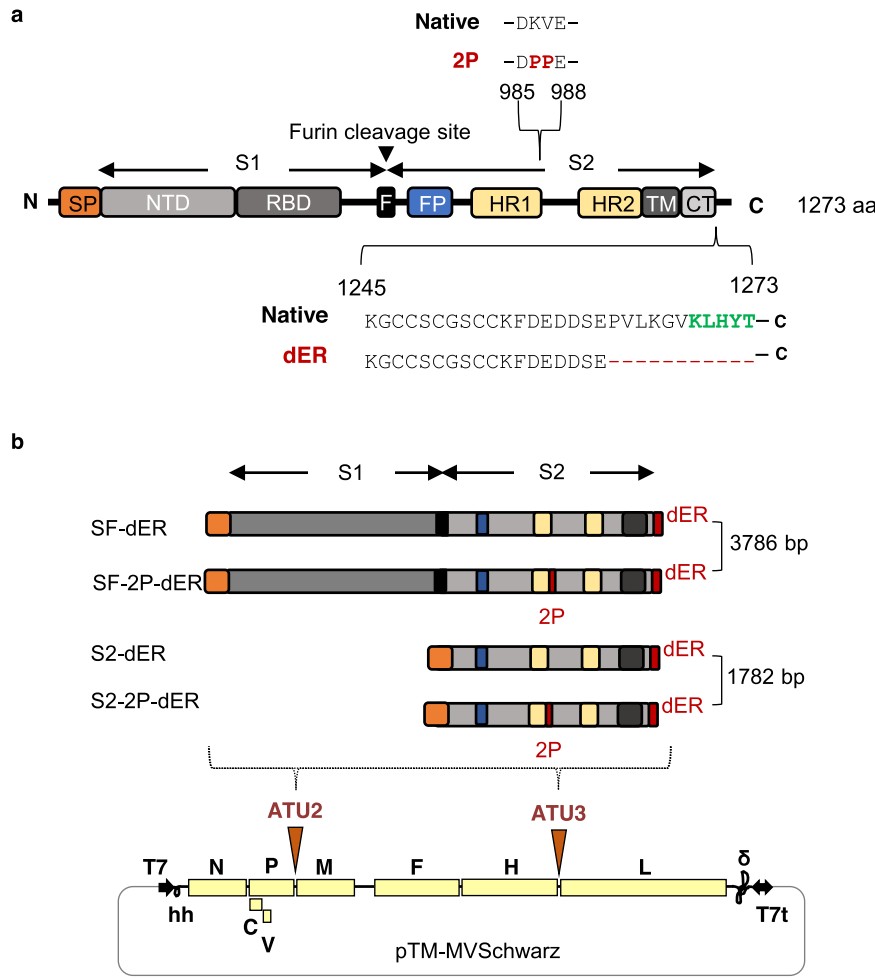

**Fig. 1 Schematic of the native S protein of SARS-CoV-2 and S gene constructs. a** The native S protein is 1273 amino acids (aa) in length. The protein contains 2 subunits, S1 and S2, generated by cleavage at the furin cleavage site (F). S1 contains the signal peptide (SP), N-terminal domain (NTD) and receptor-binding domain (RBD). S2 contains the fusion peptide (FP), heptad repeats 1 (HR1) and 2 (HR2), transmembrane domain (TM), and cytoplasmic tail (CT). The 2P indicates the two mutated prolines, K986P and V987P. The green letters indicate the endoplasmic reticulum retrieval signal (ERRS) motif, KxHxx, in the CT. dER indicates constructs carrying a deletion of the 11 C-terminal amino acids from the CT. **b** The native S gene of SARS-CoV-2 with notable domains indicated in color boxes relative to the S gene constructs cloned into the MV vector. 2P and dER modifications indicated by the red boxes. All S constructs were cloned into either the second (ATU2) or third (ATU3) additional transcription units of pTM-MVSchwarz (MV Schwarz), the MV vector plasmid. The MV genome comprises the nucleoprotein (N), phosphoprotein (P), V and C accessory proteins, matrix (M), fusion (F), hemagglutinin (H) and polymerase (L) genes. Plasmid elements include the T7 RNA polymerase promoter (T7), hammerhead ribozyme (hh), hepatitis delta virus ribozyme (∂), and T7 RNA polymerase terminator (T7t).

cells. Vero cells expressing the native S protein (full-length S with an intact CT) exhibited significant syncytium formation (Supplementary Fig. 2), indicating that functional S proteins were expressed on the cell surface. Interestingly, expression of the S2 subunit alone resulted in a hyper-fusion phenotype in Vero cells, suggesting the triggering of non-receptor-mediated membrane fusion by proteases cleaving at the S2' site and freeing the fusion peptide. In contrast, both the 2P-stabilized SF-2P-dER and S2-2P-dER did not induce syncytium formation, indicating that their fusion activity was abrogated by the 2P mutation (Supplementary Fig. 2).

**Generation of rMVs expressing SARS-CoV-2 S and S2 proteins.** The four antigenic constructs were individually cloned into the pTM-MVSchwarz plasmid in additional transcription units (ATU), with ATU2 located between the P and M genes of the MV genome and ATU3 between the H and L genes[28] (Fig. 1b). Due to the decreasing expression gradient of MV genes, cloning in ATU2 allows high-level expression of the antigen while cloning in ATU3

results in lower levels of expression[29]. Based on previous experience with MV vector, the stronger the antigen is expressed, the higher the immune responses are elicited[30]. The lower expression from ATU3 is a trade-off to facilitate rescue of rMV encoding antigens that are toxic or difficult to express.

All rMVs were successfully rescued by reverse genetics and propagated in Vero cells. Although the rMVs exhibited slightly delayed growth kinetics, final virus yields were high and identical to that of the parental MV Schwarz (~$10^7$ TCID$_{50}$/ml) (Fig. 2a). S antigen expression was detected in infected Vero cells by western blotting (WB) and immunofluorescence staining (IF) (Fig. 2b–d and Supplementary Fig. 3). As expected, much higher antigen expression was observed from ATU2 vectors compared to ATU3 (Fig. 2b). To determine whether the S protein might be incorporated onto rMV particles surface during the assembly process, we analyzed cell-free MV-ATU2-SF-2P-dER virus harvested from infected Vero cells medium. After low-speed clarification to eliminate high-density cellular debris, the medium was ultracentrifuged at high speed over a sucrose cushion, and the

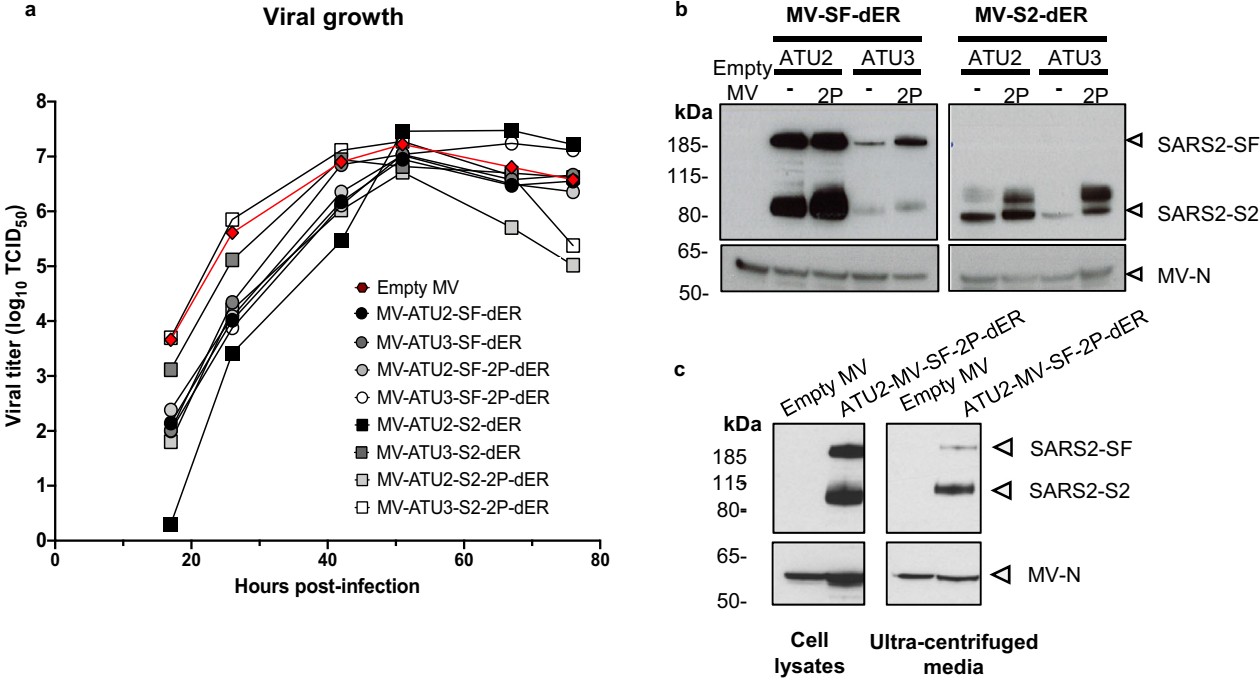

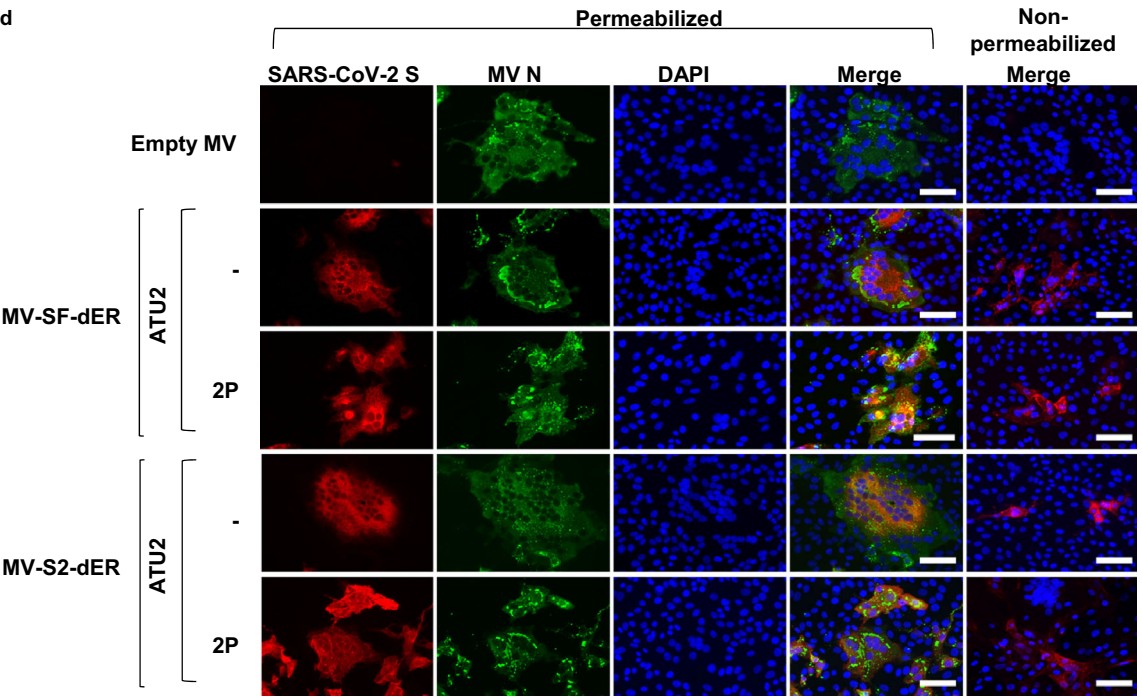

**Fig. 2 Characterization of S-expressing rMVs. a** Growth kinetics of rMV constructs used to infect Vero cells at an MOI of 0.1. Cell-associated virus titers are indicated in $TCID_{50}$/ml. **b** Western blot analysis of SARS-CoV-2 S protein in cell lysates of Vero cells infected with the rMVs expressing SF-dER or S2-dER from either ATU2 or ATU3, with or without the 2P mutation. **c** Western blot analysis of ultracentrifuged media and cell lysates of Vero infected with rMV-ATU2-SF-2P-dER. **d** Immunofluorescence staining of Vero cells infected with the indicated rMVs 24 h after infection. Permeabilized or non-permeabilized cells were stained for S (red), MV N (green) and nuclei (blue); ×20 magnification (scale bar, 50 µm). The experiments shown were conducted using two or three biologically independent Vero cell batches.

high-density material recovered from pellet was analyzed by WB[31,32] (Fig. 2c). Staining of concentrated virions with an anti-S antibody demonstrated that the S2 subunit was, at least partly, incorporated into high density material. Only a small amount of complete S was detected as it is expected to be cleaved by proteases during assembly.

The genetic stability of recombinant viral vaccines is essential to guarantee the quality of the vaccine after multiple manufacturing steps. Analysis of rMVs after serial passaging in Vero cells revealed that MV-ATU2-SF-dER, which expresses the native S from ATU2, was unstable, with loss of S expression by passage 5 (Supplementary Fig. 4A). When sequencing the cell-associated

viral RNA at passage 4 (where expression is lost), and comparing to the originally cloned sequence, we found a mixture of sequences with a lot of ADAR type mutations indicating that viral replication generated a population of mutated mRNA from which translation was not anymore possible. When analyzing by next-generation sequencing (NGS) the full-length viral RNA genome extracted from cell-free virus, we found mutations in the intergenic sequence upstream of the inserted S gene (Supplementary Fig. 4C). How these genomic mutations led to the generation of mutated mRNA remains to be determined. Hyperfusion phenotype observed with native S (Supplementary Fig. 2) was likely the selective pressure.

In contrast, MV-ATU2-SF-2P-dER counterpart was stably and efficiently expressed up to passage 10. Its sequence was also confirmed by NGS (Supplementary Fig. 5), with no mutations detected in the antigen after 10 passages of the virus. Therefore, we discarded the vaccine candidates expressing native S and selected those expressing the prefusion-stabilized SF-2P and S2-2P constructs for further immunogenicity studies.

**Induction of SARS-CoV-2 neutralizing antibodies in mice**. We investigated the immunogenicity of selected rMV vaccine candidates in IFNAR$^{-/-}$ mice, which are susceptible to MV infection[33]. Animals were immunized by one or two intraperitoneal administrations of the rMV candidates at $1 \times 10^5$ TCID$_{50}$ on days 0 and 30 (Fig. 3a). Empty MV Schwarz was used for control vaccination. Mice sera were collected 4 weeks after the prime and 12 days after the boost. The presence of S- and MV-specific IgG antibodies were assessed by indirect ELISA using SARS-CoV-2 S recombinant protein and native MV antigens, respectively.

All animals raised high MV-specific IgG antibodies after prime and boost in all groups, indicating efficient vaccine take in all the animals (Fig. 3b). Specific IgG antibodies against SARS-CoV-2 S were detected in 100% of immunized mice. Interestingly, rMVs expressing SF-2P-dER or S2-2P-dER antigens from ATU2 elicited higher levels of anti-S antibodies than the ATU3 vectors, particularly after boosting (Fig. 3c). In addition, reducing the immunization dose of the ATU2 candidate to $1 \times 10^4$ TCID$_{50}$ still induced higher NAb titers than the ATU3 vaccine at $1 \times 10^5$ TCID$_{50}$ (Supplementary Fig. 9), correlating with its higher expression level. Pre-immune sera and sera from control animals that received empty MV remained negative for anti-S antibodies (data not shown).

We next assessed the presence of SARS-CoV-2 neutralizing antibodies (NAbs) using plaque reduction neutralization tests (PRNT) with SARS-CoV-2 virus infection of Vero E6 monolayers. After the prime, SARS-CoV-2 NAbs were found in all mice immunized with SF-2P-dER expressed from ATU2 at levels similar to that of human convalescent sera, while only one mouse immunized with the ATU3 construct had NAbs (Fig. 3d). After the boost immunization, NAb titers increased in both groups, with the ATU2 group exhibited NAb titers tenfold higher compared to the ATU3 group and 40 times higher than human convalescent sera. No NAbs were detected in animals immunized with the S2 candidates despite the high levels of anti-S antibodies (Fig. 3c).

As IgG isotype switching can serve as indirect indicators of Th1 and Th2 responses[34], we determined S-specific IgG1 and IgG2a isotype titers in the sera of immunized mice (Fig. 3e, f). Similar to our previous results[12], rMV candidates elicited significantly higher IgG2a antibody titers than IgG1, reflecting a predominant Th1-type immune response (Fig. 3e, f).

**Induction of S-specific T-cell responses**. Since activated T cells play important roles in shaping Th1 and Th2 cytokine

production, we analyzed S-specific T-cell responses in MV-immunized mice in more detail. Cell-mediated immune responses were first investigated using an IFN-γ ELISPOT assay. Groups of IFNAR$^{-/-}$ mice were sacrificed one week after prime immunization (Fig. 4a). To evaluate S-specific responses, splenocytes were stimulated ex vivo with a pool of synthetic peptides covering the predicted CD8$^+$ and CD4$^+$ T-cell epitopes of the SARS-CoV-2 S protein, matching the MHC-I H-2K$^b$/H-2D$^b$ and MHC-II I-A$^b$ haplotype of 129sv IFNAR$^{-/-}$ mice (Supplementary Table 1). Splenocytes were also stimulated with an empty MV virus to detect MV vector-specific T-cell responses (negative and positive controls are presented in Supplementary Fig. 6).

High levels of T-cell responses to SARS-CoV-2 S and MV were elicited early after prime vaccination (Fig. 4a). Splenocytes from mice vaccinated with MV-ATU2-SF-2P-dER yielded high IFN-γ secretion levels after stimulation with an MHC class I-restricted S peptide pool, yielding around 2500 spot forming cells (SFC) per $10^6$ splenocytes. Lower IFN-γ responses were observed upon stimulation with MHC class II-restricted S peptides, at ~400 SFC/$10^6$ splenocytes (Fig. 4b, c). Splenocytes of these mice also exhibited relatively low vector-specific IFN-γ responses (~990 SFC/$10^6$ splenocytes), indicating a well-balanced S-to-MV vector response ratio (Fig. 4d). The ATU3 counterpart of the same vaccine tended to generate more vector-specific IFN-γ secreting cells (~1320 SFC/$10^6$ splenocytes), while at the same time being less efficient in producing S-specific IFN-γ secreting cells after stimulation with MHC class II-restricted S peptides (~130 SFC/$10^6$ splenocytes). While IFN-γ responses after stimulation with MHC class I-restricted S peptides were not significantly different from those of its ATU2 counterpart (~1980 SFC/$10^6$ splenocytes), the S-to-MV vector response ratio was significantly higher for MV-ATU2-SF-2P-dER (Fig. 4b).

In contrast, S-specific IFN-γ responses elicited by S2-only constructs were low after stimulation with either of the S peptide pools (Fig. 4c). The ratio S/MV vector response also remained low. Therefore, immunization with the S2 protein subunit alone was not sufficient to induce strong cellular immune responses (Fig. 4d). MV-ATU3-S2-2P-dER and empty MV were unable to induce S-specific IFN-γ responses.

We next studied S-specific CD4$^+$ and CD8$^+$ T cells by flow cytometric analysis after intracellular cytokine staining (ICS). S-specific IFN-γ$^+$ and TNF-α$^+$ responses were observed in CD8$^+$ T cells, while CD4$^+$ T cells responded poorly to S peptide pool stimulation (Fig. 5a, b). Similar to the ELISPOT results, SF-2P-dER expressed from ATU2 or ATU3 induced high and comparable percentages of S-specific IFN-γ$^+$ and TNF-α$^+$ CD8$^+$ T cells, while the S2 protein expressed from ATU2 was ten times less immunogenic. MV-ATU3-S2-2P-dER and empty MV were unable to induce S-specific IFN-γ- or TNF-α-producing T-cells. IL-5-secreting cells (indicative of a Th2-biased response) were not detected in any of the immunization groups. An additional detailed analysis of T cell responses in mice immunized with MV-ATU2-SF-2P-dER confirmed the strong stimulation of the CD8 compartment with high levels of S-specific IFN-γ$^+$ and TNF-α$^+$ producing CD8$^+$ T cells, as well as double-positive IFN-γ$^+$/TNF-α$^+$ producing CD8$^+$ T cells. No IL-5 or IL-13 was detected in CD4$^+$ or CD8$^+$ T cells, as well as in CD4$^+$/CD44$^+$/CD62L$^-$ memory T cells, confirming that S-specific memory T cells are also Th1-oriented (Supplementary Fig. 8).

Taken together, these results demonstrate that MV-ATU2-SF-2P-dER induces a robust Th1-driven T-cell immune response to SARS-CoV-2 S antigens at significantly higher levels than MV-ATU3-SF-2P-dER. The S2 candidates elicited much lower cellular responses and no NAb, indicating that S2 alone is not sufficient to

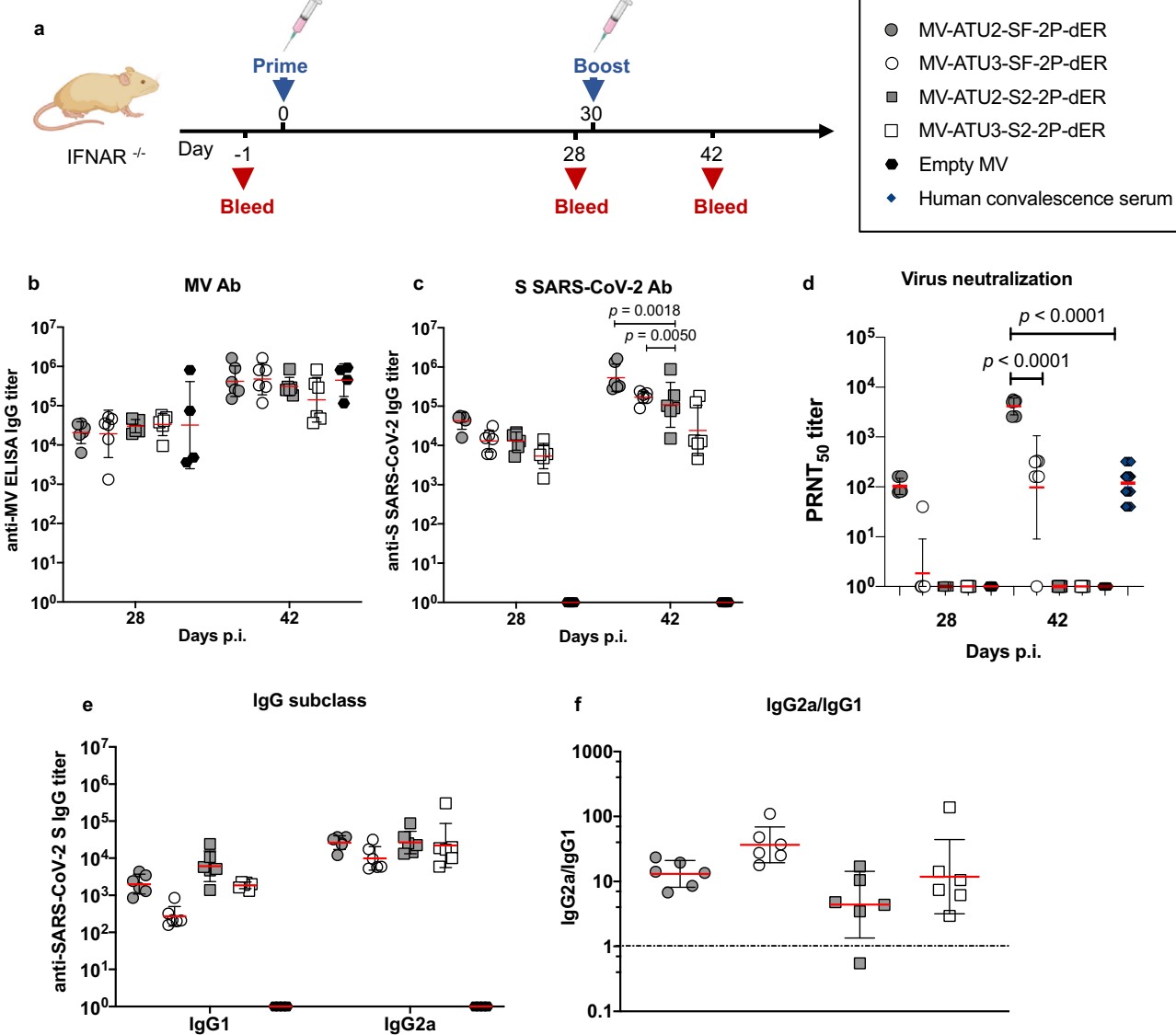

**Fig. 3 Induction of humoral responses by prime-boost vaccination. a** Homologous prime-boost of IFNAR$^{-/-}$ mice ($n = 6$ or $n = 4$ for the empty MV control) immunized intraperitoneally with $1 \times 10^5$ TCID$_{50}$ of the indicated rMV at days 0 and 28. Sera were collected 28 and 42 days after immunization and assessed for specific antibody responses to **b** MV antigens or **c** S-SARS-CoV-2 S. The data show the reciprocal endpoint dilution titers with each data point representing an individual animal. **d** Neutralizing antibody responses to SARS-CoV-2 virus expressed as 50% plaque reduction neutralization test (PRNT$_{50}$) titers. **e** IgG subclass of S-specific antibody responses in mice 4 weeks after the first immunization. **f** Ratio of IgG2a/IgG1 responses. Data are represented as geometric mean with line and error bars indicating ±geometric SD. Statistical significance was determined by two-way ANOVA with Tukey's multiple comparisons test.

induce an efficient immune response in these mice. We, therefore, excluded the S2 candidates from further analysis.

**Persistence of neutralizing antibodies and protection from intranasal challenge**. We monitored the persistence of anti-S antibodies in mice immunized twice with either MV-ATU2-SF-2P-dER or MV-ATU3-2F-2P-dER (Fig. 6a). As usually observed for MV responses (Fig. 6b), S-specific IgG titers persisted and stabilized at high levels ($10^5$–$10^6$ limiting dilution titers) for both ATU2 and ATU3 candidates for up to three months after boosting (Fig. 6c). However, immunization with the ATU2 construct resulted in significantly higher levels of S-specific IgG and NAb titers over the duration of the experiment (Fig. 6c, d).

To determine whether these responses confer protection from SARS-CoV-2 infection, immunized mice were challenged

intranasally with $1.5 \times 10^5$ PFU of MACo3, a mouse-adapted SARS-CoV-2 virus that carries mutations in RBD (S:Q493R and S:Q498R)[35]. Three days after the challenge, mice were sacrificed and the presence of the virus was examined in lung homogenates. SARS-CoV-2 RNA was measured by RT-qPCR using *RdRP* gene-specific primers[36] (Supplementary Table 2), and infectious virus levels were titered on Vero E6 cells. Although SARS-CoV-2 viral RNA was detected in the lungs of all immunized mice after the challenge, an average 2 log$_{10}$ reduction was observed in the ATU2 group and 1 log$_{10}$ reduction in the ATU3 group compared to the empty MV control group (Fig. 6e). Input virus inoculum or mRNA originating from first replication rounds may be the source of viral RNA measured in the lungs. However, no infectious virus was detected in the lungs of the ATU2 group and all but one of

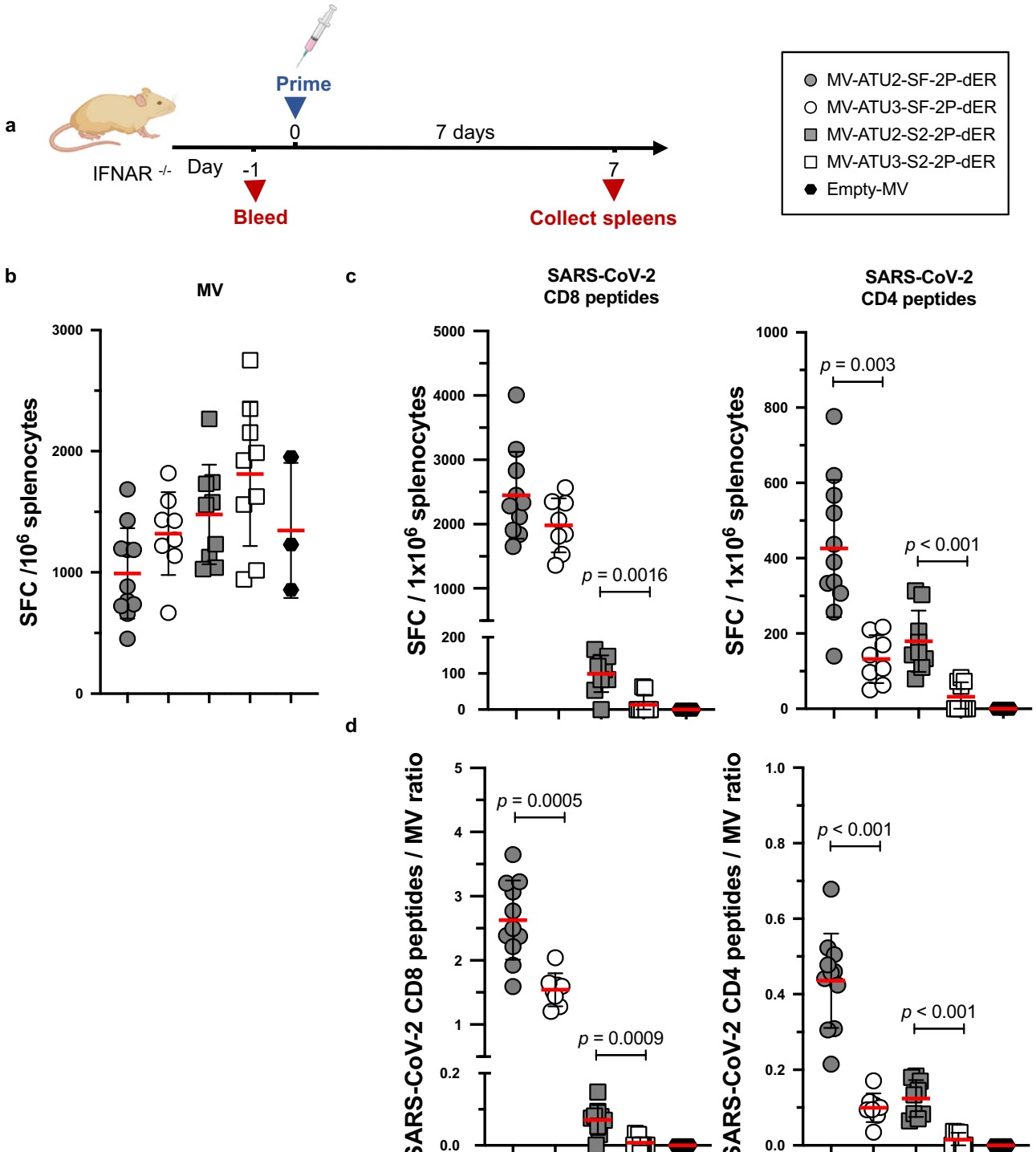

**Fig. 4 Induction of S-specific cellular responses by rMV vaccination. a** IFNAR$^{-/-}$ mice were immunized by intraperitoneal injection with $1 \times 10^5$ TCID$_{50}$ of MV-ATU2-SF-2P-dER, $n = 11$; MV-ATU3-SF-2P-dER, $n = 8$; MV-ATU2-S2-2P-dER, $n = 9$; MV-ATU3-S2-2P-dER, $n = 9$ and Empty MV, $n = 3$. Seven days after immunization, ELISPOT for IFNγ was performed on freshly extracted splenocytes. The data are shown as IFNγ-secreting cells or spot-forming cells (SFC) per $1 \times 10^6$ splenocytes detected after stimulating with **b** MV Schwarz or **c** SARS-CoV-2 S peptide pools specific to CD8$^+$ or CD4$^+$ T cells. **d** Ratio of IFNγ-secreting cells stimulated by CD4$^+$ or CD8$^+$ peptides to those stimulated by MV Schwarz. Each data point represents an individual mouse. Data are represented as mean with line and error bars indicating ±SD. Significant differences between the groups were determined by two-tailed the Mann–Whitney test.

the ATU3 group (Fig. 6f). These results demonstrate that, although viral replication may have occurred at low levels, the infectivity of the inoculated and progeny virus was efficiently neutralized.

**Partial protection from intranasal challenge after a single immunization**. We next determined whether a single immunization could protect IFNAR$^{-/-}$ mice from challenge with the MACo3 virus (Fig. 7a). Immunized animals were examined for

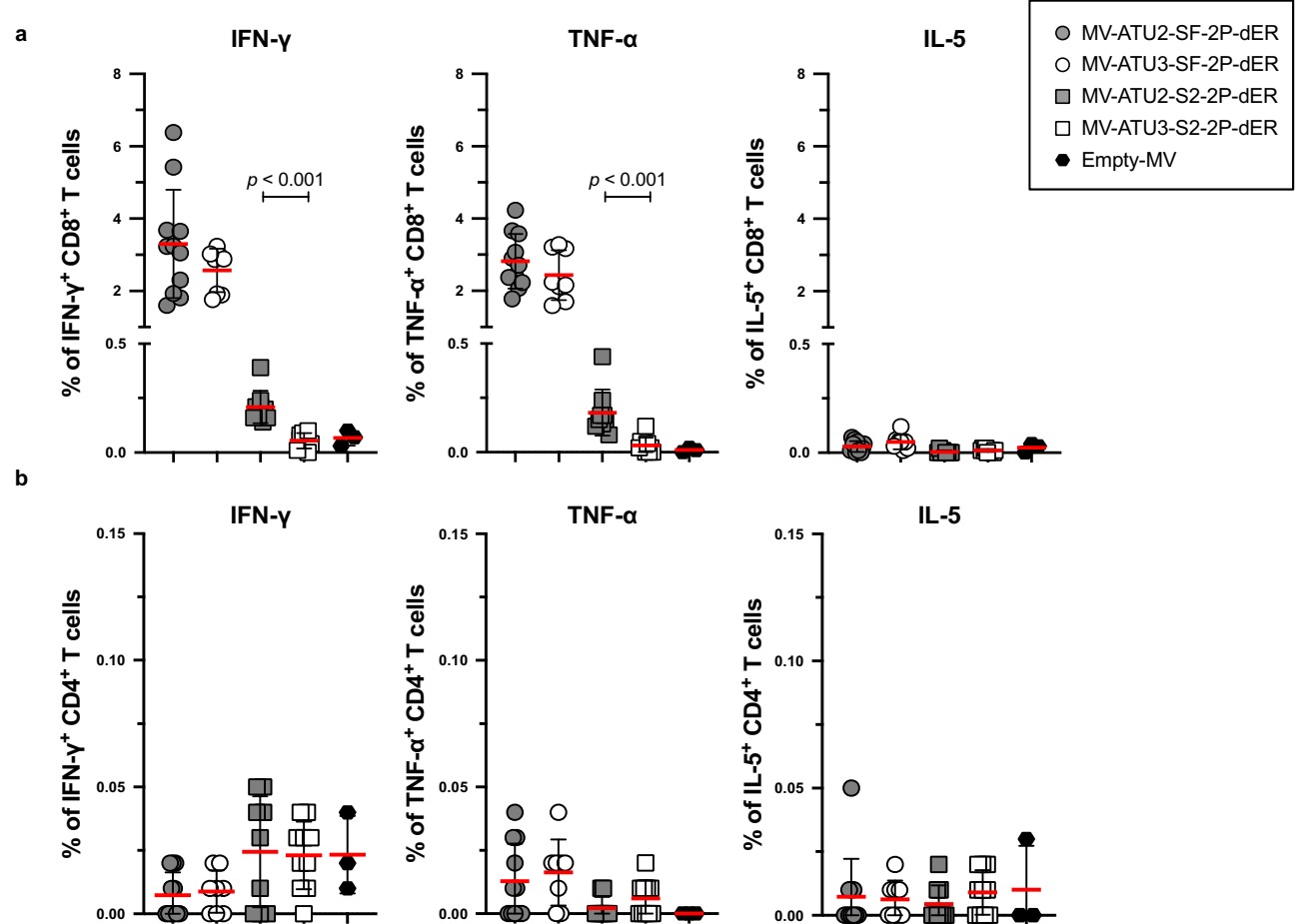

**Fig. 5 Cytokine expression profile of T cells.** IFNAR$^{-/-}$ mice were immunized by intraperitoneal injection with $1 \times 10^5$ TCID$_{50}$ of MV-ATU2-SF-2P-dER, $n = 11$; MV-ATU3-SF-2P-dER, $n = 8$; MV-ATU2-S2-2P-dER, $n = 9$; MV-ATU3-S2-2P-dER, $n = 9$ and Empty MV, $n = 3$. Seven days after immunization, splenocytes were stimulated with S-specific peptide pools. S-specific **a** CD8$^+$ and **b** CD4$^+$ T-cells were stained for intracellular IFNγ, TNFα and IL-5. Data are represented as mean with line and error bars indicating ±SD. Significant differences between the groups were determined by the two-tailed Mann–Whitney test.

immune responses on days 28 and 48 post-immunization, prior to challenge. All animals exhibited MV- and S-specific antibodies (Fig. 7b, c). Th1-associated IgG responses to the S antigen as well as SARS-CoV-2 NAbs were present before the challenge, although at lower levels than after two immunizations (Fig. 7d, e). Mice were then challenged intranasally and lung samples collected 3 days after the challenge. Although no difference was observed in viral RNA levels between the test and control groups (Fig. 7f), half of the animals immunized with MV-ATU2-SF-2P-dER were negative for infectious virus in the lungs (Fig. 7g), indicating partial protection. In contrast, animals immunized with MV-ATU3-SF-2P-dER were not protected.

**Protection of golden Syrian hamsters from intranasal SARS-CoV-2 challenge.** Using golden Syrian hamsters, a naturally permissive model of SARS-CoV-2 infection[23,37], we then tested whether a single or a prime-boost immunization with MV-ATU2-SF-2P-dER could protect the animals from intranasal challenge with a wild-type strain of SARS-CoV-2 (BetaCoV/France/IDF00372/2020) (Fig. 8a). Hamsters were immunized at day 0 and/or 21 and groups of eight animals either primed only or prime-boosted were challenged at day 35 together with a control group that received a placebo immunization with empty MV. Animals that received a prime-boost immunization with MV-ATU2-SF-2P-dER presented a limited body weight loss within the first two days post-challenge, with stabilization or body

weight gain afterwards ($p = 0.0002$ at day 4; Fig. 8b). Furthermore, these animals behaved clinically healthy during the post-challenge period, with a low clinical score compared with the placebo-immunized (empty MV) animals afterwards ($p = 0.0002$ at day 4; Fig. 8c). Animals that received a single immunization with MV-ATU2-SF-2P-dER presented an intermediate clinical picture in comparison to the prime-boost and the placebo-immunized groups (Fig. 8b, c).

Hamsters immunized with a prime-boost of MV-ATU2-SF-2P-dER presented a 3 log$_{10}$ reduction in genomic SARS-CoV-2 RNA load in the lungs ($p = 0.0002$; Fig. 8d), which corresponds to undetectable (5 animals out of 8), or low (3 out of 8) infectious viral titer in the lungs ($p = 0.0002$; Fig. 8e). Likewise, these animals presented a reduction of viral load and viral titer in the nasal turbinates (Fig. 8d, e). Animals that received a single immunization also presented reduced viral RNA loads and viral titers in both lungs and nasal turbinates, albeit less notably (Fig. 8d, e).

All hamsters immunized with two doses of MV-ATU2-SF-2P-dER, challenged or not, presented significantly higher NAbs titers than human convalescent sera ($p < 0.05$, $p < 0.01$) (Fig. 8f). All animals immunized with a single dose presented high NAbs in their serum, except for one, although at a slightly lower level than animals immunized with two doses (Fig. 8f). The Geometric mean titers (GMT) of NAbs elicited by vaccination was rapidly increased by day 4 post-challenge, indicating the efficient

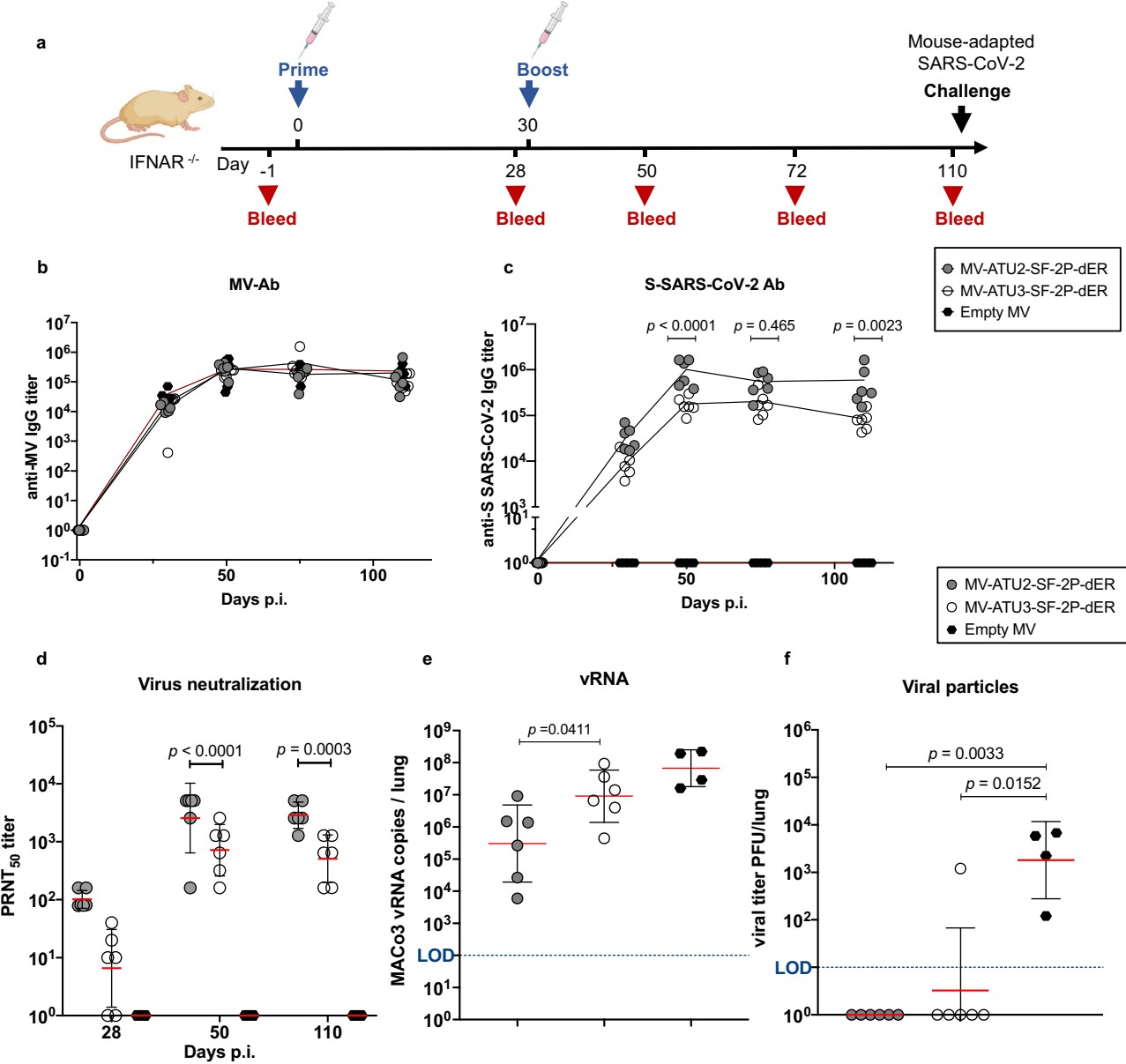

**Fig. 6 Persistence of neutralizing antibodies and immune protection. a** Immunization and challenge schedule for IFNAR−/− mice ($n = 6$ and $n = 4$ in the control group; Empty MV). Animals were immunized interperitoneally by homologous prime-boost at days 0 and 28. Sera were collected at days 52, 72, and 110. Animals were challenged on day 110 by intranasal inoculation of mouse-adapted SARS-CoV-2 virus (MACo3) at $1.5 \times 10^5$ PFU. Sera were assessed for levels of specific antibodies against **b** MV and **c** SARS-CoV-2 S. **d** Neutralizing antibody responses against SARS-CoV-2 virus, expressed as 50% plaque reduction neutralization test (PRNT$_{50}$) titers. **e** SARS-CoV-2 viral RNA copies detected by RT-qPCR in homogenized lungs of challenged animals, calculated as copies/lung. **f** Titer of infectious viral particles recovered from the homogenized lung of the immunized animals expressed as PFU/lung. Dotted blue line indicates the limit of detection (LOD). Data are represented as geometric means with line and error bars indicating ±geometric SD. Statistical significance was determined by (**c**, **d**) a two-way ANOVA with Tukey's multiple comparisons test and (**e**) two-tailed the Mann–Whitney test (**f**) Kruskal–Wallis one-way ANOVA test with Dunn's multiple comparison test.

establishment of immune memory. Interestingly, the antibodies induced by immunization with MV-ATU2-SF-2P-dER also neutralized three of the most prevalent SARS-CoV-2 variants, namely B.1.1.7 (UK variant), P.1 (Brazilian variant) and B.1.351 (South African variant) (Fig. 8g).

Finally, prime-boost immunization with MV-ATU2-SF-2P-dER protected the challenged animals from lung pathology. No or mild macroscopic changes were observed in the lungs of these animals (Supplementary Fig. 10a), which corresponds with lower lung weights (Supplementary Fig. 10b) and the absence of histopathological changes (Fig. 9). Lung histological sections of

vaccinated animals appeared healthy with no sign of pathological changes and preserved bronchiolar epithelium (Fig. 9a–d). In contrast, substantial pulmonary lesions were observed in the placebo group vaccinated with empty MV, including severe parenchyma inflammation, consolidation of pulmonary parenchyma, and marked alteration of the bronchiolar epithelium (Fig. 9a, b). Immunohistochemistry to detect SARS-CoV-2 antigens in lung tissue was negative in hamsters vaccinated with prime-boost, while control animals exhibited large numbers of SARS-CoV-2 positive cells (Fig. 9c, d). Importantly, a single dose of MV-ATU2-SF2P-dER did not abrogate lung pathology, but

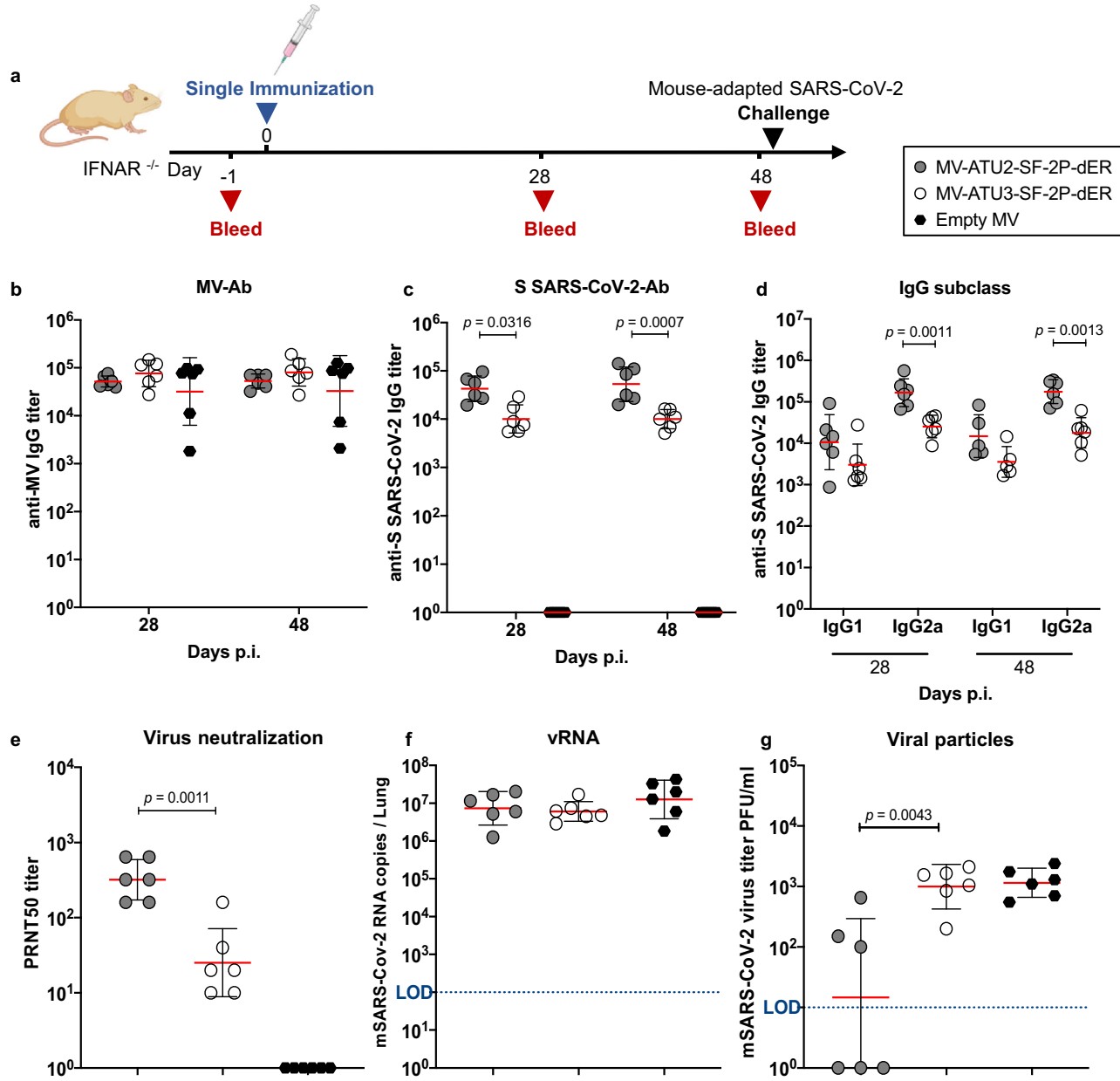

**Fig. 7 Immune responses and protection after a single immunization. a** Immunization and challenge schedule for IFNAR$^{-/-}$ mice ($n = 6$). Animals were immunized interperitoneally on day 0. Sera were collected at days 24 and 48. Animals were challenged on day 48 by intranasal inoculation of mouse-adapted SARS-CoV-2 virus (MACo3) at $1.5 \times 10^5$ PFU. Sera were assessed for levels of specific antibodies to **b** MV and **c** S-SARS-CoV-2 protein. **d** Neutralizing antibody responses against SARS-CoV-2 virus, expressed as 50% plaque reduction neutralization test (PRNT$_{50}$) titers. **e** SARS-CoV-2 viral RNA copies detected by RT-qPCR in homogenized lungs of challenged animals, calculated as copies/lung. **f** Titer of infectious viral particles recovered from the homogenized lung of the immunized animals expressed as PFU/lung. Dotted blue line indicates the limit of detection (LOD). Data are represented as geometric means with line and error bars indicating ±geometric SD. Statistical significance was determined by **b**–**d** two-way ANOVA with Tukey's multiple comparisons test, **e**–**g** Kruskal–Wallis one-way ANOVA test with Dunn's multiple comparison test.

pulmonary lesions were clearly less severe than in placebo-vaccinated animals (Fig. 9a–d).

## Discussion

In this work, we generated and tested MV-based COVID-19 vaccine candidates targeting the SARS-CoV-2 S protein. Similar to other vaccine platforms, the full-length prefusion-stabilized S was the most immunogenic, eliciting the strongest humoral and cellular responses. Our lead candidate MV-ATU2-SF-2P-dER exhibited high viral titers comparable to the wild-type Schwarz MV strain and was genetically stable up to 10 passages. In

contrast, a previously reported rMV expressing the native S from the same strong early promoter showed impaired growth[38]. One explanation could be that the 2P mutation enables a stable S protein expression by reducing the hyperfusogenic phenotype, thus allowing rMV to replicate better. MV-ATU2-SF-2P-dER elicited strong Th1-oriented T-cell responses and high titers of neutralizing antibodies to SARS-CoV-2 that persisted for months at levels around 40 times higher than human convalescent sera. Long-lasting immunity is a hallmark of live replicating vaccines[39]. T-cell responses, essential to controlling and reducing viral load and viral spread[40], were

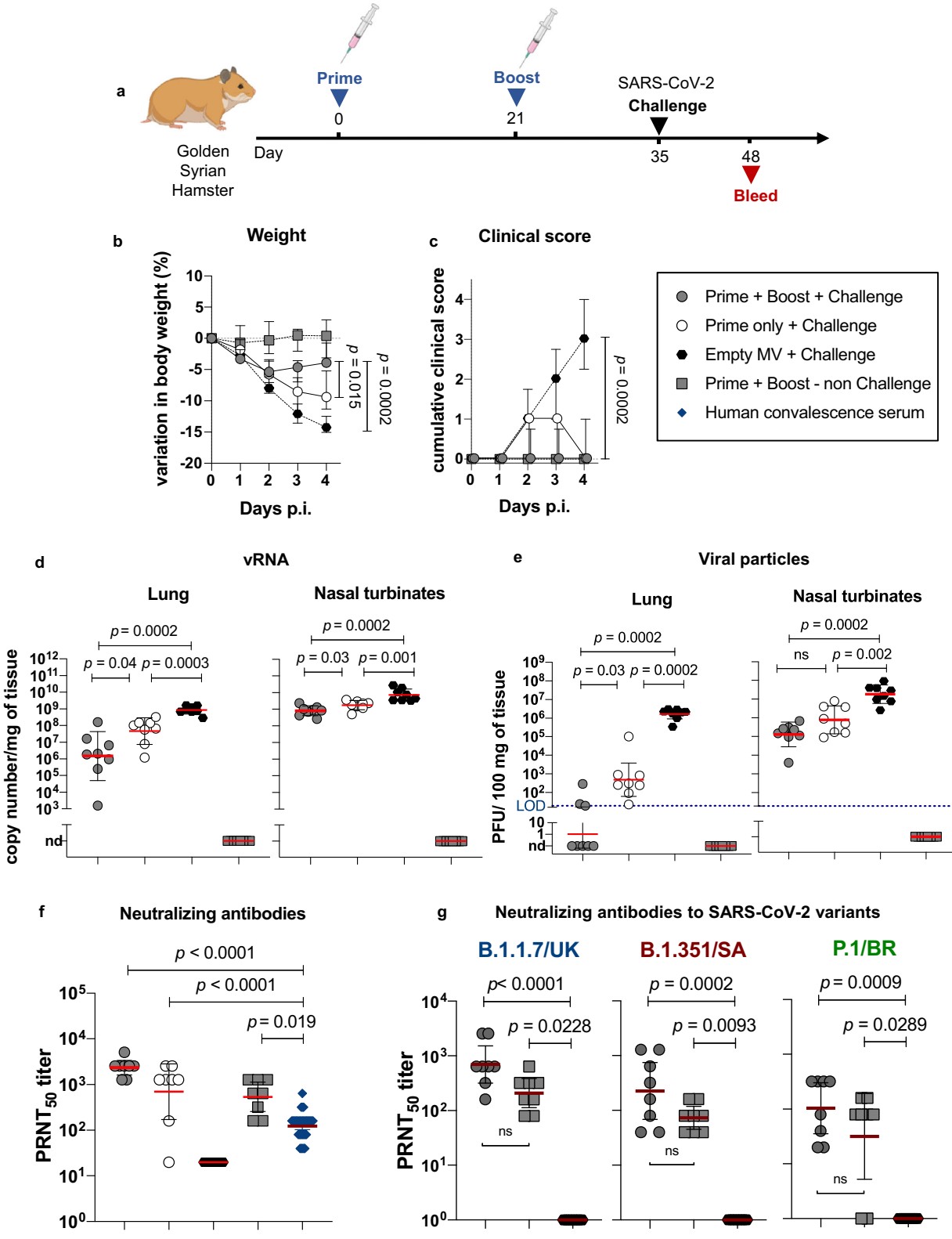

induced within seven days after a single immunization. The Th1 dominance suggests that this vaccine is less likely to induce immunopathology due to vaccine-induced disease enhancement as previously reported for SARS-CoV-1 and MERS-CoV vaccine studies[41–43]. Prime-boost immunization with MV-ATU2-SF-2P-dER afforded protection from intranasal challenge with a mouse-adapted SARS-CoV-2 virus. A single immunization

allowed sufficient immune protection according to WHO recommendations for a COVID-19 vaccine primary efficacy of at least 50%[44]. Interestingly, the mouse-adapted virus used for challenging the mice carries mutations in the spike and notably the RBD (S:Q493R and S:Q498R)[35]. This supports the idea that our vaccine candidate is able to cross protect from divergent strains of SARS-CoV-2.

**Fig. 8 MV-ATU2-SF-2P-dER immunization efficacy in golden Syrian hamsters. a** Immunization and challenge schedule for hamsters ($n = 8$/group). **b** Variation in the body weight of immunized and challenged hamsters. **c** Clinical score in immunized and challenged hamsters. The clinical score is based on a cumulative 0–4 scale: ruffled fur; slow movements; apathy; absence of exploration activity. **d** SARS-CoV-2 viral RNA copies detected by RT-qPCR at 4 dpi in homogenized lungs and nasal turbinates expressed as copy number/mg of tissue. **e** Titer of infectious viral particles recovered at 4 dpi from the homogenized lungs and nasal turbinates of the immunized animals expressed as PFU/100 mg of tissue. **f** Neutralizing antibody responses against SARS-CoV-2 virus, expressed as 50% plaque reduction neutralization test ($PRNT_{50}$) titers. **g** Neutralizing antibodies against the UK (B.1.1.7), South African (B.1.351), and Brazilian (P.1) SARS-CoV-2 variants. Dotted blue line indicates the limit of detection (LOD). Nd: not detected. Data are represented as geometric means with line and error bars indicating ±geometric SD. Statistical significance was determined by Kruskal–Wallis one-way ANOVA test with Dunn's multiple comparison test.

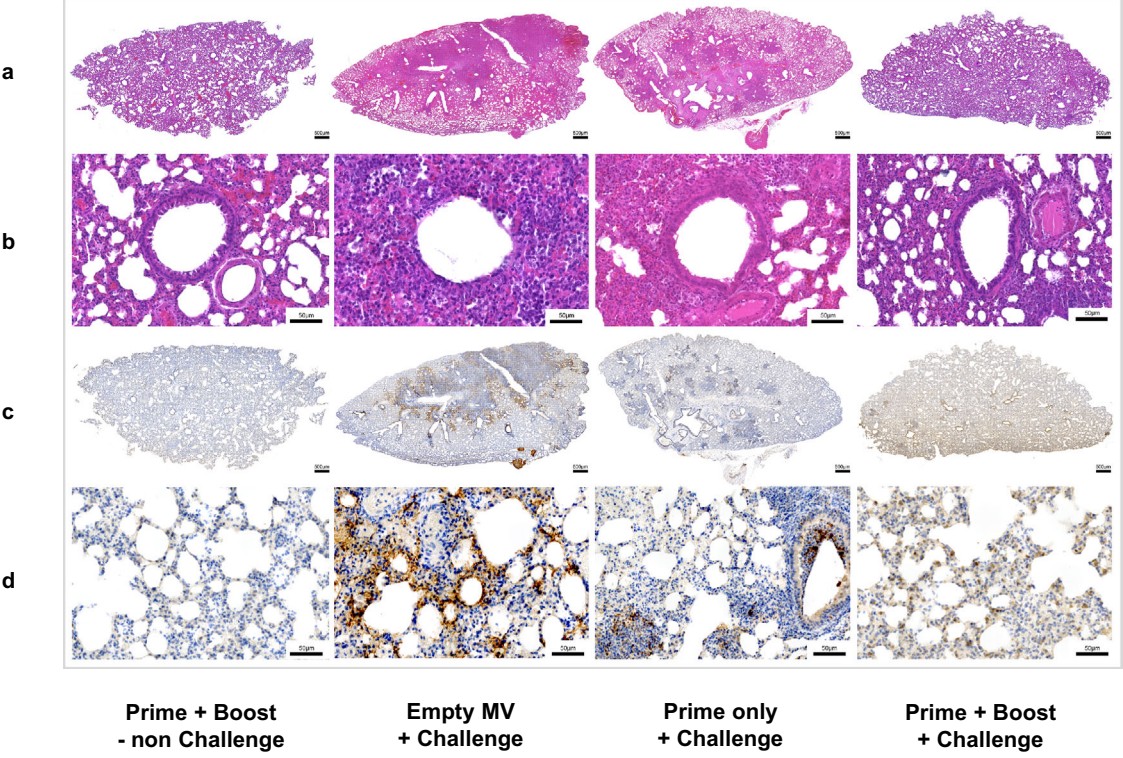

|  | Prime + Boost - non Challenge | Empty MV + Challenge | Prime only + Challenge | Prime + Boost + Challenge |

**Fig. 9 MV-ATU2-SF-2P-dER immunization protects SARS-CoV-2-challenged golden Syrian hamsters from lung pathology.** Histology and Immunohistochemistry of freshly collected lungs at 4 dpi from immunized and challenged hamsters (representative images of $n = 8$/group). Scale bars are embedded in the lower right corner of each image. **a** Hematoxylin and Eosin (H&E) stained whole-lung sections. **b** Bronchiolar epithelium sections. **c** Whole-lung sections immuno-stained with SARS-CoV-2 N antibody. **d** High magnification of the immuno-stained lung sections.

Prime-boost immunization with MV-ATU2-SF-2P-dER also protected golden Syrian hamsters from intranasal challenge with a wild-type strain of SARS-CoV-2. Vaccinated animals were protected from clinical signs and their lungs contained no infectious virus and a 1000-fold reduction in genomic SARS-CoV-2 RNA load at 4 days after challenge. No histopathological change was observed in the lungs of vaccinated animals as opposed to the placebo group. Vaccination elicited NAbs at levels 10 to 100 times higher than human convalescent sera. Notably, antibodies from the vaccinated hamsters also neutralized three of the most important current SARS-CoV-2 variants (UK-B.1.1.7, Brazilian-P.1, and South African-B.1.351). The UK variant was neutralized at titers similar to the original virus, while the African and Brazilian variants were neutralized at lower titers, although still similar to those of human convalescent sera tested against the current virus. This suggests that our lead vaccine candidate might possibly protect against a variety of SARS-CoV-2 infections and disease, whether caused by the original virus or other circulating variants, although this has still to be demonstrated in hamsters challenged with the variant viruses. Such broad-spectrum capacity of NAbs elicited by vaccination with MV-ATU2-SF-2P-dER

may result from exposure of hidden epitopes of the membrane-anchored full-length S antigen during its natural processing by a replicating virus life cycle, or from a better affinity maturation of antibodies to the pre-fusion conformation. Further studies are needed to understand the great capacity of rMV vaccine in the hamster model, particularly identifying the target cells of this live vaccine in hamster and analyzing their capacity to replicate MV in comparison with human monocytes or epithelial cells, the primary target cells of MV.

To explore another possibility of generating a broad-spectrum vaccine, we also tested a vaccine candidate expressing only the S2 subunit, which is highly conserved among SARS-CoV-1 and SARS-CoV-2 viruses. The S2 subunit has been shown to harbor immunodominant and neutralizing epitopes[16,45,46]. In this report, while S2 in the MV context induced high S-specific antibody titers, these antibodies could not neutralize the SARS-CoV-2 virus. In terms of cellular responses, S2 induced weak S-specific CD4[+] and CD8[+] T-cell responses in mice. These observations suggest that the S2 subunit alone might be insufficient for inducing immune protection. Given the low NAb titers elicited by these candidates, we did not test their protective capacity.

Our results also yielded interesting differences in the immunogenicity of rMV vaccines expressing the target antigen from ATU2 versus ATU3. S antigen is expressed at higher levels from ATU2, and this correlated with higher humoral responses, as previously reported for other recombinant MV[30]. Interestingly, higher cellular responses were also elicited to the S antigen when it was expressed from ATU2. This observation was confirmed by reducing the immunization dose of the ATU2 candidate to $1 \times 10^4$ TCID$_{50}$ that still induced higher NAb titers than the ATU3 vaccine at $1 \times 10^5$ TCID$_{50}$ (Supplementary Fig. 9). In addition, T-cell responses to the MV vector were also lower with ATU2 constructs. These observations suggest that higher antigen expression could be reducing virus replication in vivo, resulting in lower cellular responses to the vector itself. While MV vaccines have been shown to be effective despite pre-existing immunity to the vector, this more desirable balance in the immunogenicity of antigen and vector likely contributes to greater vaccine efficacy of the ATU2 construct. Nevertheless, as an rMV vehicle for future vaccines, the ATU3 concept is still useful for expressing antigens that are unstable, toxic, or otherwise difficult to express.

During the preparation of this manuscript, another MV-based COVID-19 vaccine candidate was published by Lu et al.[47]. The authors used a secreted form of the 2P-stabilized S antigen with deletion of the S1/S2 furin cleavage site and a self-trimerizing T4 fibritin trimerization motif replacing its TM and CT domains. They showed that in the context of MV expression, this antigen is immunogenic and protective in different rodent models. Deleting the furin cleavage site between S1 and S2 to generate a soluble uncleaved trimeric S molecule was necessary in their case as, otherwise, the secreted S antigen would be cleaved during its processing and presented to B cells as a trimeric S2 and a soluble S1, both of which poorly elicit NAbs as separate antigens. A secreted antigen favors an external presentation to B cells rather than to CD4 and CD8 T cells. In our case, the membrane-anchored S antigen is expressed from within cells infected by rMV and processed similarly to its native maturation during virus replication with furin cleavage active. Infected cells in vivo should present this antigen directly to both T and B cells. Indeed, although the ELISPOT and ICS protocols used were slightly different, the cellular responses measured in mice by Lu et al. were much lower than in our study using similar animals. The secreted trimeric S antigen of Lu et al. was efficiently presented to B cells and elicited NAb, but appeared poorly presented for T cells responses. Eliciting strong T-cell responses after the first dose is crucial for maintaining long-term memory. Besides, Lu et al. did not report the neutralization of emergent variants, which is now an essential feature for next-generation COVID-19 vaccines. On the regulatory aspect, the Schwarz strain MV backbone has been previously introduced several times in clinical trials after regulatory approval, while Lu et al. construct is based on an Edmonston B backbone sequence with no previous clinical data. Lastly, to recover recombinant MVs, Lu et al. used transfection into HEp-2 cells infected with a recombinant modified vaccinia Ankara virus (MVA) expressing T7 RNA polymerase. This complicates the regulatory approval pathway as viral seed needs to be MVA-free and HEp-2 cells need to be qualified for the human vaccine manufacturing process.

Despite emergency approval of several COVID-19 vaccines based on mRNA and non-replicating adenovirus vectors, additional vaccine strategies are still needed, particularly vaccines targeting the younger population and the poorest countries, and those with long duration of protection, reliable history of safety, and demonstrable efficacy against emerging variants. In those terms, MV-vectored vaccines have numerous advantages that argue for their development. First, as a replicating vector, MV-based vaccine can be administered at doses as low as $1 \times 10^5$ TCID$_{50}$, compared to adenovirus-vectored vaccines that are generally administered at $10^{10}$–$10^{11}$ viral particles per dose, running the risk of strong complement activation[48]. Some rare but severe and even deadly adverse events have been recently observed in routine vaccination with adenovirus vectors[49]. This has aroused the mistrust of several European countries towards these vaccines. The lower doses relative to production titers render a live vaccine more cost-effective for large-scale manufacturing as well. In addition, replicating vectors are known to provide long-lasting immunity[50–52]. Immune responses to our MV-ATU2-SF-2P-dER candidate lasted up to 4 months after prime-boost immunization with no sign of decline, reflecting results seen for the MV-based CHIKV vaccine[18]. MV vector naturally activates innate immunity and possesses self-adjuvanting properties[53], therefore vaccination with live attenuated rMV does not require adjuvants. Common adjuvants such as alum have been characterized to stimulate Th2-biased responses[34], which could increase the possibility of immunopathology. On a logistical aspect, the formulation and distribution of MV-vectored vaccines is much simpler than inactivated or mRNA vaccines. Recombinant MV vaccines can be distributed with the existing cold-chain infrastructure, while mRNA vaccines require ultra-cold temperatures that are much more challenging. Lastly, while adenovirus-vectored vaccines have to overcome strong pre-existing immunity to the vector, previous clinical studies demonstrated that immune responses to MV-CHIKV were not dampened when all volunteers were pre-immune to measles[18,19]. MV-CHIKV has shown excellent safety and tolerability in both phase I and phase II clinical trials. The success of MV-CHIKV thus far reaffirms that the MV vector is an excellent platform for vaccine development. Interestingly, the MV vaccine is also known for its non-specific effects (NSEs) in reducing children's mortality due to other viral infections[54,55].

In summary, we generated a replicating MV-based SARS-CoV-2 vaccine candidate that elicits in preclinical relevant rodent models strong Th1-oriented T cell responses early after priming, and high titers of NAb that persist and cross-react with new circulating variants. These immune responses result in total protection of animals from lung pathology and infectious challenge virus dissemination. These promising pre-clinical data support the development of this candidate. The constructs described in this study are different from the candidate currently being tested in clinical trials (clinicaltrials.gov identifiers NCT04497298 and NCT04498247). The numerous advantages of the MV platform argue for continuing testing alternate candidates that should help fighting against the persistent threat of COVID-19. The long safety track record of the MV along with its familiarity due to global deployment should help its acceptance as a new vaccine.

## Methods

**Cells and viruses.** Human embryonic kidney cells (HEK) 293T cells (ATCC CRL-3216), HEK293T7-NP helper cells (stably expressing MV-N and MV-P genes), African green monkey kidney cells (Vero) and Vero C1008 clone E6 (ATCC CRL-1586) were maintained at 37 °C, 5% CO$_2$ in Dulbecco's modified Eagle medium (DMEM) (Thermo Fisher) supplemented with 5% or 10% heat-inactivated fetal bovine serum (FBS) (Corning) and 1% penicillin/streptomycin (Thermo Fisher). The SARS-CoV-2 BetaCoV/France/IDF0372/2020 strain was supplied by the National Reference Centre for Respiratory Viruses hosted by Institut Pasteur (Paris, France). The human sample from which strain BetaCoV/France/IDF0372/2020 was isolated, was provided by the Bichat Hospital, Paris, France. SARS-CoV-2 variants (B.1.1.7, P.1 and B.1.351) were cultivated and amplified by the Lyssavirus, Epidemiology and Neuropathology Unit at Institut Pasteur (Paris, France). The Mouse-adapted SARS-CoV-2 (MACo-3) is described in Montagutelli et al.[35].

**Construction of pTM-MVSchwarz expressing modified SARS-CoV-2 S protein constructs.** The SARS-CoV-2 spike (S) gene, NC_045512[8], was codon-optimized for expression in mammalian cells. Primers introducing restriction sites BsiWI and

BssHII to the S 5′ and 3′ ends, respectively, were used to amplify nucleotides 1-3799 to generate full-length S (SF) with a deletion of its 11 C-terminal amino acids (SF-dER) (Fig. 1a) for cloning into the pcDNA5.1 mammalian expression vector. To generate S2 constructs, primers were designed for inverted PCR with BsmBI restriction sites and 4-nucleotide overlaps at the C-terminus of the native S signal peptide and S2 immediately adjacent to the furin cleavage site (Supplementary Table 2). The amplification product, comprising the S2 region, the pcDNA backbone, and the S signal peptide, was digested with BsmBI (NEB) and self-ligated to generate S2-dER. To maintain the conformation of S in the prefusion state, two mutations were introduced at the hinge of HR1, K986P, and V987P (2P mutation) (Fig. 1a). Primers introducing the mutations were designed with mutated overlapping nucleotides and BsmBI sites (Supplementary Table 2). The SF-dER and S2-dER constructs were amplified, digested and self-ligated to create the prefusion-stabilized SF-2P-dER and S2-2P-dER constructs. The pcDNA plasmids containing the S sequences (3 μg) were transfected into Vero cells using FugeneHD. Transfected cells were observed at 24- and 48-h post-transfection for fusogenic phenotypes.

All the inserted S genes were modified at the stop codon to ensure that the total number of nucleotides is a multiple of six[56] and were subsequently cloned into pTM-MVSchwarz encoding infectious MV cDNA corresponding to the antigenome of the MV Schwarz vaccine strain.

**Virus rescue, propagation, and titration.** Rescue of recombinant viruses was performed using a helper-cell-based system.[28] Briefly, Helper HEK293-T7-NP cells were transfected with 5 μg of pTM-S-SARS-CoV2 and 0.02 μg of pEMC-La expressing the MV polymerase L gene. After overnight incubation at 37 °C, the transfection medium was replaced by fresh medium and a heat shock was applied for 3 h at 42 °C and then returned to 37 °C. After two days of incubation at 37 °C, transfected cells were transferred to 100-mm dishes with monolayers of Vero-NK cells (ATCC, CCL-81). Syncytia that appeared after 2–3 days of co-culture were singly picked and transferred onto Vero cells seeded in six-well plates. Infected cells were trypsinized and expanded in 75-cm$^2$ and then 150-cm$^2$ flasks, in DMEM with 5% FBS. To collect viruses, cells were scraped into a small volume of OptiMEM (Thermo Fisher), lysed by a single freeze-thaw cycle and cell lysates clarified by low-speed centrifugation. The infectious supernatant was then collected and stored at −80 °C. Titers of rMVs were determined on Vero cells seeded in 96-well plates infected with serial tenfold dilutions of virus in DMEM with 5% FBS. After incubation for 7 days, cells were stained with crystal violet, and TCID$_{50}$ values were calculated using the Karber method.[57] Titers of SARS-CoV-2 and MACo3 were assessed on Vero-E6 in a similar plaque assay. Plaques were counted 3 days post-infection. Virus growth kinetics of rMVs was studied on monolayers of Vero cells in six-well plates. Cells were infected with rMVs at an MOI of 0.1. At various time points post-infection, infected cells were scraped into 1 ml OptiMEM, lysed by freeze-thaw, clarified by centrifugation, and titered as described above.

To assess the stability of S antigen expression by recombinant viruses, Vero cells were repeatedly infected for ten passages. Virus was collected by a freeze-thaw cycle after 1, 5, and 10 passages (P1, P5, P10) and used to infect Vero cells in six-well plates in duplicate. Cell lysates were then assessed for S mRNA and protein levels using RT-PCR, western blotting, and NGS respectively.

**RT-PCR.** To verify S expression from the rMV constructs, total RNA were extracted from infected Vero cells using the RNeasy Mini Kit (Qiagen). The cDNA synthesis and PCR steps were performed using the RNA LA PCR kit (Takara Bio) with primers targeting ATU2 and ATU3, according to the manufacturer's instructions. RT-PCR products were verified by Sanger sequencing (Eurofins Genomics). All primers used are indicated in Supplementary Table 2.

**Sucrose gradient ultracentrifugation.** Vero cells in T-150 flasks were infected with rMVs at MOI of 0.1. Supernatants collected after 48 h post-infection were clarified at 300 × g for 10 min, then layered onto a 20% sucrose cushion in PBS and ultra-centrifuged at 160,000 × g for 3 h in a SW41 rotor. Pellets were resuspended in PBS with protease inhibitors (Roche) and analyzed by SDS-PAGE and immunoblotting described below.

**Western blot analysis.** Vero cells in six-well plates were infected with various rMVs at an MOI of 0.1. At 36–48 h post-infection, infected cells were lysed in RIPA lysis buffer (Thermo Fisher). Samples were briefly centrifuged and subjected to 4–12% gradient NUPAGE-PAGE gel (Invitrogen). After transfer to a nitrocellulose membrane (GE Healthcare), the membrane was subsequently probed with a rabbit polyclonal anti-SARS-CoV S antibody recognizing the conserved 1124 aa–1140 aa epitope (ABIN199984, Antibody Online, 1:2000 dilution) followed by a horse-radish peroxidase (HRP)-conjugated swine anti-rabbit IgG antibody (P0399, Dako, 1:3000 dilution). Bands were visualized using SuperSignal West Pico Plus chemiluminescent HRP substrate (Thermo Fisher). For loading controls, membranes were stripped with 5% NaOH for 5 min and re-probed with a mouse monoclonal anti-MV-N antibody (ab9397, Abcam, 1:20,000 dilution) followed by an HRP-conjugated anti-mouse IgG (NA931V, GE Healthcare, 1:10,000 dilution).

**Next-generation sequencing.** Extracted RNA was treated with Turbo DNase (Ambion) followed by purification using SPRI beads (Agencourt RNA clean XP, Beckman Coulter). We used a ribosomal RNA (rRNA) depletion approach based on RNAse H and targeting human rRNA. The RNA from the selective depletion was used for cDNA synthesis using SuperScript IV (Invitrogen) and random primers, followed by second-strand synthesis. Libraries were prepared using a Nextera XT kit and sequenced on an Illumina NextSeq500 (2 × 150 cycles) at the Mutualized Platform for Microbiology hosted at Institut Pasteur. Raw reads were trimmed using Trimmomatic v0.39[58] to remove adaptors and low-quality reads. We assembled reads using metaSPAdes v3.14.0[59] with default parameters. Scaffolds were queried against the NCBI non-redundant protein database (13) using DIAMOND v2.0.4 (14). No other viruses were detected. The recombinant MV genomes identified were verified and corrected by iterative mapping using CLC Assembly Cell v5.1.0 (QIAGEN). Aligned reads were manually inspected using Geneious prime v2020.1.2 (2020) (https://www.geneious.com/), and consensus sequences were generated using a minimum of 5× read-depth coverage to make a base call. Minor variants frequencies were estimated using Ivar[60].

**Immunofluorescence assay.** Vero cells were infected with various rMVs at MOI of 0.1. At 24–36 h post-infection, cells were fixed with 4% paraformaldehyde, blocked with 2% goat serum overnight, and then treated with or without 0.1% saponin A (Sigma). Fixed cells were probed with a mouse monoclonal anti-SARS-CoV S antibody (ab273433, Abcam, 1:300 dilution) and followed by Alexa Fluor 488-conjugated goat anti-rabbit IgG (A-11008, Thermo Fisher). Staining with anti-MV-N followed by Cy3-conjugated goat anti-rabbit (A10520, Jackson ImmunoResearch, 1:1000 dilution) was used to detect MV in the same infected cells. Nuclei were stained with DAPI. Images were collected using an inverted Leica DM IRB fluorescence microscope with a ×20 objective.

**Flow cytometry.** The pcDNA5.1 plasmid expression vectors encoding prefusion-stabilized or native conformation full-length S and S2 subunit antigens were used to transfect HEK293T cells using the JetPrime transfection kit (PolyPlus) according to the manufacturer's instructions. Forty-eight hours post-transfection, cells were stained for indirect immunofluorescence with 10 μg/ml of rabbit polyclonal anti-S antibody targeting S2 (ABIN199984) followed by Alexa Fluor 488-conjugated goat anti-rabbit IgG (A-11008). Propidium iodide was used to exclude dead cells. Stained cells were acquired on the Attune NxT flow cytometer (Invitrogen) and data were analyzed using FlowJo v10.7 software (FlowJo LLC).

**Mice immunizations and challenge.** We previously observed that mice deficient for type-I IFN receptor (IFNAR$^{-/-}$) are susceptible to MV infection and immunization without the need for transgenic hCD46 human receptor[33]. We therefore used 129sv IFNAR$^{-/-}$ mice that have the same MHC haplotype than the classical C57BL/6 mice: MHC-I (H-2Kb/H-2Db) and MHC-II (I-Ab). Groups of 6 to 8-week-old mice were intraperitoneally (IP) injected with 10$^5$ TCID$_{50}$ rMV, namely SF-2P-dER or S2-2P-dER in ATU2 or ATU3, or the control empty MV Schwarz. To study humoral responses, two immunizations were administered at a 4-week interval. Sera were collected before the first immunization (day −1) and then before (day 28) and after (day 42) the second immunization. All serum samples were heat-inactivated for 30 min at 56 °C. To assess protection, mice that received either one or two immunizations were challenged with an intransal inoculation of 1.5 × 10$^5$ PFU mouse-adapted SARS-CoV-2 virus (MACo3). Three days after the challenge, mice were sacrificed and lung samples collected. The presence of MACo3 virus in the lung was detected by determining viral growth, PFU of infectious viral particles, and measuring vRNA using Luna Universal Probe One-Step RT-qPCR kit following the manufacturer's protocol (E3006). The primers and probes used correspond to the nCoV_IP4 panel (S.Table2) as described on the WHO website[36].

**Golden Syrian hamsters immunization and challenge.** Golden Syrian hamsters (*Mesocricetus auratus*; RjHan:AURA) aged 5-6 weeks were divided into four groups (4 females and 4 males/group); (1) (MV-ATU2-SF-2P-dER, prime+boost, challenged), (2) (MV-ATU2-SF-2P-dER prime only, challenged), (3) (empty MV, prime+boost, challenged), and (4. (MV-ATU2-SF-2P-dER prime+boost, non-challenged). Each group received IP injection of 5 × 10$^5$ TCID$_{50}$ of either MV-ATU2-SF-2P-dER or Empty MV. All, except group 2, were boosted with the same conditions at Day 28. (Fig. 8a). Fourteen days after the boost, each animal was intranasally administered 100 μL (50 μL/nostril) of physiological solution containing 6 × 10$^4$ PFU (plaque-forming unit) of SARS-CoV-2 (BetaCoV/France/IDF00372/2020)[23]. Non-challenged animals received only physiological solution. Challenged and non-challenged animals were housed in separate isolators and all hamsters were followed-up daily for four days during which the body weights and the clinical scores were noted. Clinical scores were based on a cumulative 0–4 point scale: ruffled fur, slow movements, apathy, absence of exploration activity. At day 4 post-challenge, animals were euthanized[61]. Blood samples were collected by cardiac puncture. After coagulation and centrifugation, sera were collected and frozen at −80 °C until use. Samples of nasal turbinates and lungs were collected and frozen at −80 °C. Fragments of lungs were also collected, fixed in 10% neutral-buffered formalin, and embedded in paraffin. Four-μm-thick

sections were cut and stained to describe histological lesions (hematoxylin–eosin staining), or labeled to assess SARS-CoV-2 distribution using anti-N SARS-CoV (NB100-56576, NovusBio). All histological procedures were performed with the Bond III automat (Leica Biosystems).

**Viral titration and RNA isolation from golden hamsters' lungs and nasal turbinates**. Frozen lungs and nasal turbinates fragments from the golden Syrian hamsters were weighted and homogenized with 1 mL of ice-cold DMEM (Gibco) supplemented with 1% penicillin/streptomycin (Thermo Fisher) in Lysing Matrix M 2 mL tubes (MP Biomedicals) using the FastPrep-24™ system (MP Biomedicals).

For viral titration, the tissue homogenate supernatants were titrated on Vero-E6 cells by classical plaque assays using semisolid overlays (Avicel, RC581-NFDR080I, DuPont) and expressed per PFU/100 mg of tissue[62].

For RNA isolation, the tissue homogenate supernatants were mixed with Trizol LS (Invitrogen) and the total RNA from nasal turbinates was extracted using the Direct-zol RNA MicroPrep Kit (Zymo Research) and from lungs with the (Zymo Research). The presence of SARS-CoV-2 RNA in these samples was evaluated by Superscript III Platinum One-Step RT-qPCR (Invitrogen) containing the nCoV_IP2 and the nCoV_IP2 probe (S.Table 2) according to the manufacturer's protocol. Viral load quantification (RNA copy number/mg of tissue) was assessed by linear regression using a standard curve of RNA transcripts containing the $RdRp$ sequence (ranging from $10^7$ to $10^2$ copies).

**ELISA**. Edmonston strain-derived MV antigens (Jena Bioscience) or recombinant S protein (ABIN6952426, Antibodies Online) were coated on NUNC MAXISORP 96-well immuno-plates (Thermo Fisher) at 1 µg/ml. Coated plates were incubated overnight at 4 °C. After washing and blocking, sera from immunized mice were serially diluted in the binding buffer and incubated on plates for 1 h at 37 °C. After washing steps, an HRP-conjugated goat anti-mouse IgG (H + L) antibody (Jackson ImmunoResearch, 115-035-146, 1:5000 dilution) was added for 1 h at 37 °C. Antibody binding was detected by the addition of the TMB substrate (Eurobio) and the reaction was stopped with 100 µl of 30% $H_2SO_4$. The optical densities were recorded at 450 and 620 nm wavelengths using the EnSpire 2300 Multilabel Plate Reader (Perkin Elmer). Endpoint titers for each individual serum sample were calculated as the reciprocal of the last dilution giving twice the absorbance of the negative control sera. Isotype determination of the antibody responses was performed using HRP-conjugated isotype-specific (IgG1 or IgG2a) goat anti-mouse antibodies (AB97240 and AB97245, Abcam, 1:5000).

**Plaque reduction neutralization test**. Two-fold serial dilutions of heat-inactivated serum samples were incubated at 37 °C for 1 h with 50 PFU of SARS-CoV-2 virus (wild-type or variants) in DMEM medium without FBS and added to a monolayer of Vero E6 cells seeded in 24-well plates. The virus was allowed to adsorb for 2 h at 37 °C. The supernatant was removed and the cells were overlaid with 1 ml of plaque assay overlay media (DMEM supplemented with 5% FBS and 1.5% carboxymethylcellulose). Plates were incubated at 37 °C with 5% $CO_2$ for 3 days. Viruses were inactivated and cells fixed and stained with a 30% crystal violet solution containing 20% ethanol and 10% formaldehyde (all from Sigma). Serum neutralization titer was counted on the dilution that reduced SARS-CoV-2 plaques by 50% ($PRNT_{50}$).

**Human convalescent patients sera**. Human sera from convalescent patients were supplied by Armed Forces Biomedical Research Institute (IRBA) from the cohort study IMMUNO-COVID-PERCY approved by a research ethics committee. The main initial goal of this cohort was to follow the length of the serological and cellular immune response among health care workers after one of the first important cluster in an hospital setting during the first wave of COVID-19 in France. All patients were informed and provided a written consent to use their serum.

**ELISPOT**. Splenocytes from immunized mice were isolated and red blood cells lysed using Hybri-Max Red Blood Cell Lysing Buffer (Sigma). Multiscreen-HA 96-well plates (Millipore) were coated overnight at 4 °C with 100 µl per well of 10 µg/ml of anti-mouse IFN-γ (551216, BD Biosciences) in PBS. Plates were washed and blocked with supplemented complete MEM-α media (Thermo Fisher). The medium was replaced with 100 µl of cell suspension containing $1 \times 10^5$ splenocytes per well in triplicate and 100 µl of stimulating agent in complete MEM-α supplemented with 10 U/ml of mouse IL-2 (Roche). Stimulating agents used were 2.5 µg/ml concanavalin A (Sigma Aldrich) for positive controls, complete MEM-α for negative controls, MV Schwarz virus at an MOI of 1, or a SARS-CoV-2 S peptide pool (Supplementary Table 1) at 2 µg/ml per peptide. After incubation for 40 h at 37 °C, 5% $CO_2$, plates were washed once with PBS, then three times with washing buffer (PBS, 0.05% Tween). Biotinylated anti-mouse IFN-γ antibody (554410, BD Biosciences) at 1 µg/ml in wash buffer was added and plates were incubated for 120 min at room temperature. After extensive washing, 100 µl of streptavidin–alkaline phosphatase conjugate (Roche) was added at a dilution of 1:1000. After 1-hour incubation at room temperature, wells were washed twice with the wash buffer and followed by a wash with PBS buffer without Tween. Spots were developed with BCIP/NBT (Sigma) and counted on a CTL ImmunoSpot®

ELISPOT reader. Data with negative and positive controls are presented in Supplementary Fig. 6.

**Intracellular cytokine staining**. Splenocytes of vaccinated mice were extracted as described above. Two million splenocytes per mouse per well were incubated in 200 µL of complete MEM-α medium (Thermo Fisher). BD Golgi Stop (554724, BD Biosciences) was added to the culture medium according to the manufacturer's instructions. Splenocytes were stimulated with a peptide pool covering predicted CD4 and CD8 T-cell epitopes of the SARS-CoV-2 S protein (Supplementary Table 2) at a final concentration of 2 µg/ml per peptide. PMA/Ionomycin Cell Stimulation Cocktail (eBioscience) was used as a stimulation for positive controls, and medium alone was used for negative controls. Splenocytes were stimulated for 4 h at 37 °C. Stimulated cells were incubated with Mouse BD Fc Block (553141, BD Biosciences), and stained with Live/Dead Fixable Aqua Viability Dye (Thermo-Fisher) to exclude dead cells by gating. Subsequently, cells were stained with CD3e PE (clone 145-2C11, 12-0031-83, eBioscience), CD4 PerCP-eFluor710 (clone RM4-5, 46-0042-82) and CD8 Alexa Fluor 488 (clone 53-6.7, 53-0081-82) antibodies from Invitrogen. Cells were fixed and permeabilized with BD Fixation/Permeabilization kit (BD Biosciences) and stained with 1:200 dilution of IFN-γ APC/Fire750, (clone XMG1.2, 505860, BioLegend), TNF-α BV421 (clone MP6-XT22, 563387, BD Horizon), IL-5 APC (clone TRFK5, 505860, BD Biosciences), IL-13 eFluor 660 (clone eBio13A, 50-7133-82, eBioscience), CD62L APC-Cy7 (clone MEL-14, 104427, BioLegend), and CD44 BV786 (clone IM7, 563736, BD Pharmingen) antibodies. Samples were acquired using the Attune NxT flow cytometer (Invitrogen) and data were analyzed using FlowJo v10.7 software (FlowJo LLC). Gating strategy information with positive and negative controls is presented in Supplementary Fig. 7.

**Statistical information**. Statistical analyses were performed using GraphPad Prism v.9.0. Results were considered significant if $p < 0.05$. The lines in all graphs represent in mean values with error bars indication ±mean SD or geometric mean values with error bars indicating ±geometric SD. Statistical analyses of antibody responses, ELISA and $PRNT_{50}$, were done using two-way ANOVA adjusted for multiple comparisons. The two-tailed nonparametric Mann–Whitney $U$ test was applied to compare differences between two groups. Statistical analyses of challenged animals, weights, vRNA, viral particles, and clinical score, were done using the one-way ANOVA Kruskal–Wallis test.

**Ethics**. All animal experiments were performed according to French legislation in compliance with the European Communities Council Directives (2010/63/UE, French Law 2013–118, 6 February 2013) and according to the regulations of Institut Pasteur Animal Care Committees. The Animal Experimentation Ethics Committee (CETEA 89) of the Institut Pasteur approved this study (200023) before experiments were initiated. The animals were manipulated in class III safety cabinets in the Institut Pasteur animal facilities accredited by the French Ministry of Agriculture for performing experiments on live rodents. All animals were handled in strict accordance with good animal practice. Animals were maintained on a 12 h light/dark cycle at 18–24 °C and minimum humidity of 40%.

**Reporting summary**. Further information on research design is available in the Nature Research Reporting Summary linked to this article.

## Data availability

Source data are provided as a Source Data file. The source data supporting the findings of this study have also been deposited in a public repository database (10.6084/m9.figshare.14959518) and a direct link can be obtained upon request to the corresponding author. Source data are provided with this paper.

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

## Acknowledgements

Funding for "Outbreak Response to Novel Coronavirus (COVID-19): Development of a vaccine against novel coronavirus SARS-CoV-2 using the clinical Phase III measles vector technology" project was provided by the Coalition for Epidemic Preparedness Innovations (CEPI). The development of the mouse-adapted MacO3 virus was funded by the ANR-20-COVI-0028-01 grant program. P.N.F. was supported by the ANR-18-CE17-0004-01 programme, Recherche translationnelle en santé. Part of this work was supported by Institut Pasteur TASK FORCE SARS COV2 (NicoSARS and NeuroCovid projects). We thank Anastassia Komarova, Vincent Enouf and Sylvie Van Der Werf from the RNA Viruses Molecular Genetics Unit of Institut Pasteur for helpful discussions and SARS-Cov-2 virus information. We thank Maud Vanpeene and Vincent Enouf from the Mutualized Platform of Microbiology, Pasteur International Bioresources Network for viral sequencing. We thank Victoire Perraud and Simon Bonas from the Lyssavirus Epidemiology and Neuropathology Unit for their help with animal experiments and RNA extraction. We thank Audrey Ferrier, Annabelle Garnier, and Laurence Cheutin from the Armed Forces Biomedical Research Institute (IRBA) for handling the serums cohort from SARS-Cov-2 convalescent patients. The human sample from which strain BetaCoV/France/IDF0372/2020 was isolated was provided by Drs. Xavier Lescure, Yazdan Yazdanpanah from the Bichat Hospital, Paris. The human sample from which strain hCoV-19/France/ IDF-IPP00078/2021 (B.1.1.7, UK variant) was isolated was provided by Dr Mounira Smati-Lafarge, from CHI de Créteil, France. The human sample from which strain hCoV-19/France/IDF-IPP11324/2020 (B.1.351, South African variant) was isolated was provided by Dr Foissaud, HIA Percy, France. These strains were supplied by the National Reference Centre for Respiratory Viruses hosted by Institut Pasteur (Paris, France) and headed by Dr. Sylvie van der Werf. The human sample from which strain hCoV-19/Japan/TY7-501/2021 (P,1, Brazilian variant, JPN [P.1]) was supplied by the Japanese National Institute of Infectious Diseases (Tokyo, Japan). SARS-CoV-2 variants (B.1.1.7, B.1.351, and P.1) were cultivated and amplified by Unit Lyssavirus, Epidemiology and Neuropathology at Institut Pasteur (Paris, France) headed by Hervé Bourhy.

## Author contributions

P.N.F., A.B., A.J., G.D.M., E.S.L., H.B., and F.T. conceived the study; P.N.F., A.B., C.R., C.C., V.N., M.P., L.P., X.M., P.F., H.S.M., G.D.M., F.L., L.C., L.K., D.H., M.T., L.L., and E.S.L. performed experiments; P.N.F., A.B., H.S.M., J.D.S., E.S.L., G.D.M., E.M.B., J.N.T., and F.T. analyzed data; P.N.F., A.B., and G.D.M. performed statistical analysis; P.N.F., A.B., S.T. and F.T. wrote the manuscript.

## Competing interests

F.T. and P.N.F. are inventors of a patent describing the vaccine constructs and filed by Institut Pasteur. All other authors declare no competing interests.
