## [Peer Review File · Nature Communications]

A live measles-vectored COVID-19 vaccine induces strong immunity and protection from SARS-CoV-2 challenge in mice and hamstersReviewers' Comments:

Reviewer #1:

Remarks to the Author:

Frantz et al. describe in their manuscript „A measles-vectored COVID-19 vaccine induced long-term immunity and protection from SARS-CoV-2 challenge in mice ” the generation and analysis of a COVID-19 vaccine candidate based on recombinant Schwarz vaccine-strain derived MV encoding different versions of the SARS-CoV-2 Spike glycoprotein as additional antigen.

The authors generate different versions of the CoV Spike glycoprotein with modifications to enhance surface expression, to stabilize the antigen structure, and to direct immune responses to conserved regions of the protein. They demonstrate that the recombinant MV with such an additional antigen can be rescued and that some of these constructs are genetically stable. In IFNAR-ko mice, the genetically stable constructs induce humoral and cellular immune responses directed against MV and SARS-CoV-2. While anti-S antibodies became induced by all constructs, SARS-CoV-2 neutralization became evident only for constructs encoding both subunits of S, while the candidates encoding S2 induced no neutralizing response. The distribution of antibody isotypes indicated a Th1-biased immunity, a finding which was confirmed by cytokines induced in T cells of immunized mice after re-stimulation with S-peptide. T cell responses were of considerable extent and multifunctional Th1-biased S-specific T cells were observed. Finally, immunized mice challenged with a mouse-adapted SARS-CoV-2 revealed abrogation of live SARS-CoV-2 and significant reduction of SARS-CoV-2 genome copy numbers up to 2 log₁₀ in the lungs after intranasal challenge after prime-boost immunization, while single-shot vaccination was not as effective but nevertheless significantly reduced live virus titers in the respectively immunized cohort.

The authors propose that their data provide evidence for efficacy of measles vaccine-based COVID-19 vaccine constructs that should be pushed through clinical trials also for vaccine logistics and economic reasons.

Albeit very well designed and controlled experiments were performed, and their results are of high interest for the field, the manuscript should profit from considering the following points:

Major points:

1.) While the authors provide convincing, well-controlled data identifying an optimized version of S as a superior antigen to be included into recombinant MV, and their data concerning the S2 antigen reveal very valuable insights into future direction of COVID-19 vaccine development especially in the face of the recently up-coming SARS-CoV-2 field variants with mutated, partially neutralization-resistant Spike glycoproteins, they do not cite the recent paper by Hörner et al. and set their data into perspective. Comparing the novel data to the already published ones which relied on a recombinant MV encoding unmodified SARS-CoV-2 S will further indicate the effects of using an optimized antigen format besides corroborating the reproducibility and validity of the MV vaccine platform system.

2.) Rapid deletion of the antigen expression in MV-ATU2-SF-dER, but not in the construct with the stabilized S-version seems somewhat surprising to the reviewer with a view on the comparable cell-associated growth of both viruses according to Fig. 2b. Evolutionary selection pressure should be comparable for both, according to the latter. Differences e.g. in generation of free vaccine virus particles could be an explanation, and determination of virus titers in the supernatant over time should be performed to strengthen or dismiss this hypothesis, while yielding better insight into the vaccine virus growth characteristics which are critical for vaccine production, in addition.

3.) The authors suggest that their vaccine design suppresses (“minimizing”, abstract l. 34) anti-vector immunity in the favor of immunity against the antigen on the basis of significantly enhanced T cell responses against S in some of their constructs and a weak tendency for lower anti-MV responses in the same constructs expressing high amounts of the antigen. That immunity against the additional antigen encoded by recombinant MV can be boosted by enhancing relative expression of the

respective antigen in infected cells has been shown before for antibody responses using Hepatitis B virus antigen (HBsAg) (del Valle et al. 2007). That this enhancement also applies for T cell responses is for sure an important information, which should be discussed in this background. However, the reviewer does not agree that the preferable, significant enhanced induction of anti-S responses concurrently down-modulates anti-MV responses. Data in Figure 4 are too weak to significantly support this conclusion, while no strict correlation is found, e.g. MV-ATU2-S2-2P-dER virus induces higher anti-CoV-S T cells, but also higher anti-MeV T cells when compared to MV-ATU3-SF-2P-dER.

Minor points:

1. Limits of detection of the assays should be indicated on the graphs, where appropriate. E.g. Fig.6f suggests that a single infectious virus per ml lung homogenate could have been detected, Fig. 6e a single viral genome per lung, which for technical reasons seems rather unlikely.
2. Fig. 2b: Graph is overcrowded and symbols are too small to delineate the time-course of individual constructs – the manuscript would profit from splitting the data in two graphs.
3. Supplementary Figures S3 and S4 are neither described nor discussed in the text in some detail, while their content seems to be at least partially redundant to Fig. 2c. The paper would profit from removal of redundancies or more detailed description.
4. For T cell assays, data for positive (ConA, PMA/Iono) and negative controls of stimulation (as mentioned in the materials and methods section) should be included in the paper
5. Supplementary Figure SFig.8 is not described in the results section
6. L. 40: Please indicate date of “to date”
7. L. 104 ff.: The authors may want to outline the rationale why they choose to increase surface expression of S and their proposed mechanism of enhanced induction of which immune responses.
8. L. 222 f; l. 240: In the absence of challenge data and correlates of protection, the reviewer does not fully agree with the strong conclusion/suggestion, that vaccine candidates encoding a version of S2 will not induce strong “protective” cellular immune responses. Of course, the constructs with full S induce stronger and protective immunity and the conclusion to go ahead with those is fully justified, but on the basis of current data, protective efficacy of the S2-encoding MV vaccines cannot be ruled out.
9. L. 249: Unit of indicated titers does not seem to be appropriate: “limiting dilution” titer indicating Nab titer measured by PRNT50?
10. L. 259: How do the authors rule out input virus inoculum as the source of genome copies measured in lungs of vaccinated mice, but must have been generated by viral replication?
11. L. 370: for clarity, GenBank acc.no. of SARS-CoV-2 sequences could be indicated
12. L. 374 et al.: Plasmid name most likely contains typo: pcDNA instead of pCDNA ?
13. L. 383: Please indicate amounts of plasmids transfected into Vero cells
14. L. 476: Please indicate for clarity reasons Sv129 background of IFNAR-/- mice, here.
15. L. 573: CEPI project number of research grant is missing

Reviewer #2:

Remarks to the Author:

This study analyses the efficacy of a measles vaccine strain-based vaccine expressing SARS-CoV-2 spike protein. This is an excellent and solid study using standard methods to analyze the vaccine in a mouse system. The data presented indicate the vaccine is immunogenic. However, there are no controls like human convalescence sera to compare the vaccine's immune responses to human responses, a standard method. Moreover, important experiment such as analyzing the vaccine in the presence of preexisting immunity against the measles virus is not performed.

Other significant concerns with the study are two-fold. A similar vaccine has already been published (<https://www.pnas.org/content/117/51/32657>), and this study adds little to the field of measles virus based vaccines. Recently, Merck announced that this measles-based vaccine failed in phase 1 human trial based on the lack of immunogenicity, which further reduces enthusiasm for publication in this

journal. The study is still suitable for publication in a more specialized journal such as nature reports.

Reviewer #3:

Remarks to the Author:

This manuscript describes the use of infectious measles virus as a vector to deliver the SARS-CoV2 S protein or the S2 part of the protein as a vaccine. They have inserted the S gene into the MV genome in two locations showing the expected result that insertion closer to the 3' end of the genome results in higher expression. The authors report antibody and T cell responses in mice after intra peritoneal immunization. The S protein and the S2 contained the previously characterized pre-fusion stabilizing mutations as well as deletion of the ER retrieval sequences, previously described by others, in the wild type S protein to facilitate cell surface expression. Using a mouse adapted SARS-CoV2 virus, they report that immunization with the MV-S protected mice from virus challenge.

The experiments that are reported are relatively comprehensive and adequately done. However, there are some issues that detract from the study.

1. The manuscript repeats similar experiments, reported in 2014, characterizing responses to measles virus with the SARS-CoV S protein gene inserted into the genome with apparently very similar results, a not unexpected finding since the S proteins of the two viruses are very similar. Thus, this manuscript reports results that are not particularly novel. The only compelling aspect is that the authors use the SARS-CoV2 S protein.

2. Given the recently approved COVID vaccines now in use and others in phase 3 trials, it would have been more informative had the authors compared their results with those of vaccines now in use in terms of efficacy, ease of preparation, and other advantages beyond those outlined in the discussion. Also given the failure of the Merck VSV vectored S protein as a vaccine, the authors should consider why they think their MV vector will overcome problems encountered by Merck.

3. The use of IP administration of the vaccine candidate is less than optimal. The authors should have used either IM or IN administration for results more applicable to human populations.

4. The use of a mouse adapted SARS virus is a real concern. There is no information provided about this virus or the changes that resulted in the genome to allow it to replicate in mice. Thus, how adequately it reflects the wild type virus is unknown. The authors should have used a mouse strain transgenic for ACE-2 or hamsters. Either model is more widely accepted.

5. Missing are data documenting histology of lungs of the mice after virus challenge in order to assess any potential for enhanced disease. This is important due to the history of enhanced disease upon use of some MV vaccine candidates or SARS-CoV vaccine candidates.

6. The authors argue that responses in mice to the MV-S are long term defining long term as 3 months. This is really not long enough for a good test of the durability of the responses. Longer times should be tested.

7. The authors should clarify the structure of the S protein used as target in ELISA. Apparently, this S protein used contains mutations, the rationale of which is not explained. The target S protein needs to be validated by binding to soluble ACE-2 and/or monoclonal Ab specific to the RBD of S protein or the S2.

8. There are at least two cases where the panels in the figures do not correspond to the text or the figure legends. (Figure 3 and Figure 7). This issue certainly makes reading the manuscript more

difficult. The authors should carefully check all text and figure legends for correspondence to the figures.

Reviewer #1 (Remarks to the Author):

We thank the reviewer for the valuable comments and acknowledging the importance and solidity of our work.

Major points:

1.) While the authors provide convincing, well-controlled data identifying an optimized version of S as a superior antigen to be included into recombinant MV, and their data concerning the S2 antigen reveal very valuable insights into future direction of COVID-19 vaccine development especially in the face of the recently up-coming SARS-CoV-2 field variants with mutated, partially neutralization-resistant Spike glycoproteins, they do not cite the recent paper by Hörner et al. and set their data into perspective.

At the time of writing our manuscript, the paper by Hörner et al. was only available on BioRxiv and did not went through peer-reviewed evaluation. We are now including in the revised manuscript the citation of the Hörner et al. paper but also the other paper by Lu et al. that was more recently published during the evaluation of our manuscript. We added to our manuscript a long discussion paragraph comparing the different candidates and supporting the advantages of ours.

Comparing the novel data to the already published ones which relied on a recombinant MV encoding unmodified SARS-CoV-2 S will further indicate the effects of using an optimized antigen format besides corroborating the reproducibility and validity of the MV vaccine platform system.

We fully agree with the reviewers' comment that a comparison of the immune responses induced by recombinant MV encoding unmodified spike (Hörner et al.) or a secreted modified spike (Lu et al.) with our new data are important. We thus modified our text to highlight that our improved antigen expression induces higher virus neutralization titers and T cell responses in mice than those described by Hörner et al. or Lu et al. More importantly, we extend our mice observations to experiments performed in the hamster model and now include new data and figures showing that our lead vaccine candidate is efficient at protecting hamsters challenged with native SARS-CoV-2 virus as well at eliciting antibodies that neutralize in vitro three other variants of SARS-CoV-2. (Fig 9).

2.) Rapid deletion of the antigen expression in MV-ATU2-SF-dER, but not in the construct with the stabilized S-version seems somewhat surprising to the reviewer with a view on the comparable cell-associated growth of both viruses according to Fig. 2b. Evolutionary selection pressure should be comparable for both, according to the latter. Differences e.g. in

generation of free vaccine virus particles could be an explanation, and determination of virus titers in the supernatant over time should be performed to strengthen or dismiss this hypothesis, while yielding better insight into the vaccine virus growth characteristics which are critical for vaccine production, in addition.

We did not observe deletion of the antigen, but rather the loss of its expression. When sequencing the cell-associated virus at passage 4 (where expression is lost), and comparing to the originally cloned sequence, we found a mixture of sequences with lot of mutations (ADAR type), meaning that viral replication generated a population of mutated mRNA from which translation was not anymore possible. When analyzed in single cycle growth kinetics, the virus replicated well and cell-associated titer was good because the expression of the harmful insert was knocked down. Virus titer in the supernatant was similar although lower as usual. When sequencing by NGS the full-length virus genome RNA extracted from cell-free virus, we found clusters of mutations in the intergenic sequence upstream the inserted gene. These genomic mutations are likely responsible for the generation of mutated mRNA. Concerning the phenotype, we observed that inserting native S sequence led to a hyperfusogenic phenotype that was reduced by stabilizing the spike with the 2-P mutation. Hyperfusion was likely the selective pressure but it remains to be determined.

3.) The authors suggest that their vaccine design suppresses (“minimizing”, abstract I. 34) anti-vector immunity in the favor of immunity against the antigen on the basis of significantly enhanced T cell responses against S in some of their constructs and a weak tendency for lower anti-MV responses in the same constructs expressing high amounts of the antigen. That immunity against the additional antigen encoded by recombinant MV can be boosted by enhancing relative expression of the respective antigen in infected cells has been shown before for antibody responses using Hepatitis B virus antigen (HBsAg) (del Valle et al. 2007). That this enhancement also applies for T cell responses is for sure an important information, which should be discussed in this background. However, the reviewer does not agree that the preferable, significant enhanced induction of anti-S responses concurrently down-modulates anti-MV responses. Data in Figure 4 are too weak to significantly support this conclusion, while no strict correlation is found, e.g. MV-ATU2-S2-2P-dER virus induces higher anti-CoV-S T cells, but also higher anti-MeV T cells when compared to MV-ATU3-SF-2P-dER.

We agree with the reviewer that the data presented in Fig.4 are not sufficient to demonstrate that increased anti-S responses down-modulate the anti-MV responses, particularly in ATU2 SF-2P-dER constructs. Nevertheless, eliciting increased T cell response to the inserted antigen over the vector responses is a major advantage of this specific construct. This

observation is obviously linked to differential viral replication *in vivo* that needs to be further examined. The sentence was removed from the abstract, the text modified accordingly and reference was added in the results and discussion sections (L371).

Minor points:

1. Limits of detection of the assays should be indicated on the graphs, where appropriate. E.g. Fig.6f suggests that a single infectious virus per ml lung homogenate could have been detected, Fig. 6e a single viral genome per lung, which for technical reasons seems rather unlikely.

Limits of detection to both tests were added in figures 6, 7 and 8.

2. Fig. 2b: Graph is overcrowded and symbols are too small to delineate the time-course of individual constructs – the manuscript would profit from splitting the data in two graphs.

Figures 1 and 2 were modified and separated differently to give more space and make the growth curves more easily readable.

3. Supplementary Figures S3 and S4 are neither described nor discussed in the text in some detail, while their content seems to be at least partially redundant to Fig. 2c. The paper would profit from removal of redundancies or more detailed description.

The redundancy was removed and these images are now presented in a single SFig. 3.

4. For T cell assays, data for positive (ConA, PMA/Iono) and negative controls of stimulation (as mentioned in the materials and methods section) should be included in the paper.

We have now added two new supplemental figures to show the data for T cell stimulation controls. We now indicate in Materials and Methods section that the ELISPOT data with negative and positive controls are presented in SFig. 6. Similarly, the gating strategy information with positive and negative controls for T cells stimulation in ICS experiments are now presented in SFig. 7.

5. Supplementary Figure SFig.8 is not described in the results section.

It is now S.Fig. 9 and described in the discussion section (L374-376) to emphasize the significant difference in the antigen expression from ATU2 and ATU3.

6. L. 40: Please indicate date of “to date”

The sentence was rewritten.

7. L. 104: The authors may want to outline the rationale why they choose to increase surface expression of S and their proposed mechanism of enhanced induction of which immune responses.

We have added a sentence to justify the choice for surface expression of the antigen from our previous work with SARS-CoV (L.106, L.110). In the discussion section this point is discussed from L.393.

8. L. 222 f; l. 240: In the absence of challenge data and correlates of protection, the reviewer does not fully agree with the strong conclusion/suggestion, that vaccine candidates encoding a version of S2 will not induce strong “protective” cellular immune responses. Of course, the constructs with full S induce stronger and protective immunity and the conclusion to go ahead with those is fully justified, but on the basis of current data, protective efficacy of the S2-encoding MV vaccines cannot be ruled out.

We agree with the reviewer that because the animals vaccinated with the S2 constructs were not challenged, we cannot rule out that these low responses might have been protective. The text has been slightly modified (L.225-228, L.245).

9. L. 249: Unit of indicated titers does not seem to be appropriate: “limiting dilution” titer indicating Nab titer measured by PRNT50?

The sentence was modified (L.253).

10. L. 259: How do the authors rule out input virus inoculum as the source of genome copies measured in lungs of vaccinated mice, but must have been generated by viral replication?

We agree with the reviewer that virus inoculum or first replication round may account for the detected RNA in lungs of vaccinated animals. Moreover, the qPCR used was not able to discriminate between messenger or genomic RNA. The whole paragraph has been changed to make it clearer (L.260-267).

11. L. 370: for clarity, GenBank acc.no. of SARS-CoV-2 sequences could be indicated

GenBank accession: NC_045512 was indicated. (L.470)

12. L. 374 et al.: Plasmid name most likely contains typo: pcDNA instead of pCDNA?

All have been changed to pcDNA (L.473-485)

13. L. 383: Please indicate amounts of plasmids transfected into Vero cells

The amounts of transfected plasmids have been added (L.485).

14. L. 476: Please indicate for clarity reasons Sv129 background of IFNAR^{-/-} mice, here.

This information was added (L570-573) and background is explained.

15. L. 573: CEPI project number of research grant is missing

The detail of the funding by CEPI is indicated in the form suggested by CEPI (L731-733).

Reviewer #2 (Remarks to the Author):

We thank the reviewer for his comment and for acknowledging the exceptional importance and solidity of our data.

This study analyses the efficacy of a measles vaccine strain-based vaccine expressing SARS-CoV-2 spike protein. This is an excellent and solid study using standard methods to analyze the vaccine in a mouse system. The data presented indicate the vaccine is immunogenic. However, there are no controls like human convalescence sera to compare the vaccine's immune responses to human responses, a standard method. Moreover, important experiment such as analyzing the vaccine in the presence of preexisting immunity against the measles virus is not performed.

Concerning the control with human convalescent sera, we thank the reviewer for his suggestion and apologize for forgetting this important control in our previous version. In the revised version we added a comparison of immunized mice and hamster's sera to human convalescent patients' sera, which shows that the NAb titers of immune sera are 10-40 times higher than those of human convalescent sera (Fig.3d and Fig.9).

Concerning the issue of preexisting immunity to measles vector, this is an important point that we and others addressed previously in several studies. We demonstrated that preexisting immunity in humans did not or barely affect the immunogenicity of measles vector to the additional antigen in the context of a MV-CHIKV vaccine. Phase I and II clinical trials demonstrated that the vaccine was immunogenic in 100% of volunteers despite the presence of high anti-measles preexisting immunity in all of them (Brandler et al., Vaccine 2013; Ramsauer et al, Lancet Inf. Dis. 2015; Reisinger et al, Lancet, 2018). Same observation was made in preclinical trials in mice and non-human primates with MV Lassa, MV-HIV or MV-CHIKV vaccines.

Other significant concerns with the study are two-fold. A similar vaccine has already been published (<https://www.pnas.org/content/117/51/32657>), and this study adds little to the field of measles virus based vaccines.

During the preparation of this manuscript two reports were published in PNAS describing other measles-based COVID-19 vaccine candidates (Hörner et al. & Lu et al.). We added to our revised manuscript a long discussion paragraph comparing the different candidates and supporting the advantages of ours.

Recently, Merck announced that this measles-based vaccine failed in phase 1 human trial based on the lack of immunogenicity, which further reduces enthusiasm for publication in this journal. The study is still suitable for publication in a more specialized journal such as nature reports.

The candidate tested by Merck and Institut Pasteur in the CEPI context and that demonstrated insufficient immunogenicity in clinical phase I in 2020 is not the one described in this work. This information is given in the revised manuscript (L.447). Merck is willing to restart clinical trials with an improved measles-COVID19 candidate.

Reviewer #3 (Remarks to the Author):

The experiments that are reported are relatively comprehensive and adequately done. However, there are some issues that detract from the study.

1. The manuscript repeats similar experiments, reported in 2014, characterizing responses to measles virus with the SARS-CoV S protein gene inserted into the genome with apparently very similar results, a not unexpected finding since the S proteins of the two viruses are very similar. Thus, this manuscript reports results that are not particularly novel. The only compelling aspect is that the authors use the SARS-CoV2 S protein.

We agree with the reviewer that this work is the adaptation of MV vaccine platform to COVID-19 infection. This study is indeed based on our work previously performed in 2014 on SARS-CoV1. We describe several strategic differences in the antigen design compared to our 2014 paper, we explore the influence of 2P mutation, dER, and compare full length S and S2 antigens in their cellular immunogenicity, virus neutralization and efficacy of protection. Moreover, we have now added the hamster model for SARS-CoV-2.

2. Given the recently approved COVID vaccines now in use and others in phase 3 trials, it would have been more informative had the authors compared their results with those of vaccines now in use in terms of efficacy, ease of preparation, and other advantages beyond those outlined in the discussion. Also given the failure of the Merck VSV vectored S protein as a vaccine, the authors should consider why they think their MV vector will overcome problems encountered by Merck.

Obviously, since our first submission of this manuscript, the situation has changed regarding vaccines now in use. The introduction and discussion sections of our revised manuscript have been changed accordingly. All the points raised by the reviewer are addressed in the discussion to compare the different approaches. A control with human convalescent sera has been added to all experiments that allows comparing our results with other strategies. The levels of NAb elicited in mice or hamsters in our study is 10-100 times higher than those of human convalescent sera. Comparing to the level obtained in non-human primates or in humans with the vaccines already in use, such level of NAb can be predicted as protective in humans. With the pandemic not slowing down, the spread of multiple variants and insufficient vaccine production to meet global demand, we strongly believe that a live measles-COVID19 vaccine could become part of the solution with its rapid and cheap manufacturing capacity and its cold-chain logistics already existing.

The candidate tested by Merck and Institut Pasteur in the CEPI context and that demonstrated insufficient immunogenicity in clinical phase I in 2020 is not the one described in this work. This information is given in the revised manuscript (L447). Merck is willing to restart clinical trials with an improved measles-COVID19 candidate.

3. The use of IP administration of the vaccine candidate is less than optimal. The authors should have used either IM or IN administration for results more applicable to human populations.

For convenience we generally use the IP route in our mouse model. We previously compared (in house, not published) the different routes of administration and did not observe much difference. As MV is replicating, the route of immunization does make a difference.

4. The use of a mouse adapted SARS virus is a real concern. There is no information provided about this virus or the changes that resulted in the genome to allow it to replicate in mice. Thus, how adequately it reflects the wild type virus is unknown. The authors should have used a mouse strain transgenic for ACE-2 or hamsters. Either model is more widely accepted.

We agree with the reviewer, and this is why we have now used the hamster model with a wild type strain of the virus. We note that the mouse adapted virus used carries mutations in the spike and notably the RBD (S:Q493R and S:Q498R). This supports the idea that our vaccine candidate is able to cross protect from divergent strains of SARS-CoV-2. We confirm now this capacity by showing that sera from immunized hamsters cross neutralize three current variants of SARS-Cov-2.

5. Missing are data documenting histology of lungs of the mice after virus challenge in order to assess any potential for enhanced disease. This is important due to the history of enhanced disease upon use of some MV vaccine candidates or SARS-CoV vaccine candidates.

Data documenting histopathology and histochemistry of lungs in the hamster model have been added to the revised version (Fig.9).

6. The authors argue that responses in mice to the MV-S are long term defining long term as 3 months. This is really not long enough for a good test of the durability of the responses. Longer times should be tested.

We agree with this comment. We have previously observed for other rMV candidates that the antibodies lasted up to 1 year or more in these mice (Brandler et al. 2008).

7. The authors should clarify the structure of the S protein used as target in ELISA. Apparently, this S protein used contains mutations, the rationale of which is not explained. The target S protein needs to be validated by binding to soluble ACE-2 and/or monoclonal Ab specific to the RBD of S protein or the S2.

The S protein used in the ELISA was from commercial origin (ABIN6952426, Antibodies Online). This protein contains the deltaFurin mutation (R683A, R685A). The certificate of analysis indicates that this protein binds efficiently to human ACE2 receptor.

8. There are at least two cases where the panels in the figures do not correspond to the text or the figure legends. (Figure 3 and Figure 7). This issue certainly makes reading the manuscript more difficult. The authors should carefully check all text and figure legends for correspondence to the figures.

The figures were modified and verified.

Reviewers' Comments:

Reviewer #1:

Remarks to the Author:

The Reviewer would like to thank the authors for considering the points made and the extra data provided which significantly enhance the quality and content of this manuscript. Although most points were clearly addressed, some issues may or should still be addressed:

- Considering major point 2 of the original review, the authors indicate in their point-by-point reply the explanation for the loss of antigen-expression in one of their candidates, but the data seemingly have not been incorporated into the manuscript. Since this observation is of general interest for the platform technology, the authors are strongly encouraged to incorporate this information in the final version of their manuscript.

- del Valle et al. should be acknowledged and cited in l. 111 for providing evidence that higher amounts of antigen expressed by recombinant MV triggers better antibody titers

- Suppl. Figs. 6, 9 and 10 are not described in the manuscript's result section. The enclosed information should be transmitted in the text, when necessary, or the Figs should be deleted, if not. The reviewer suggests the first option.

- Discrepancies between the result text and figure seems to become evident concerning the new hamster experiments depicted in Fig. 8:

a.) panel e: The text states that 7/8 animals had no virus titer in the lungs, but just 5 are displayed as "not detectable", whereas 2 animals had titers at the limit of detection and 1 revealed low titers. Either the 2 animals at the LOD had some remaining virus (than, this is no absence of virus titer) or they had not - than they should be displayed as "non detectable". Please clarify.

b.) panel f: The text states that 7/8 animals immunized with a single dose showed NAb titers (l. 296); in the figure, all animals had nAb titers, although one with just a titer of slightly higher than 10. Please clarify.

- Fig. 8f: Why are NAb titers of hamsters with prime-boost, but w/o challenge lower than the titer of hamster cohorts vaccinated once or twice, but later on challenged?

- Fig. 9 / l. 303ff: Please clarify, which hamster cohort exactly is meant by the term "protected animals". Fig. 9 depicts pathologic changes in the "prime only" cohort, at least.

- l. 323f.: "that the 2P mutation enables a stable S expression allowing rMV to replicate better" should be specified. Do the authors mean that a stabilized S is better for MV replication due to absence of fusion activity (as documented in the revised manuscript and elsewhere)?

- l. 335: The information of the RBD-mutations of the mouse-adapted virus should be included already in the result section.

- l. 351: link of broad-spectrum neutralization also of variants to the exposure of hidden epitopes seems quite speculative, here. Also for e.g. mRNA vaccines, degree of cross-neutralization of variants has been linked to the potency of the induced antibodies against the original virus - better affinity maturation of the antibodies in B cell clones against the pre-fusion conformation could be an exemplary alternative explanation for the observation made.

- l. 396ff.: The reviewer does not agree on significant obvious disadvantages of the constructs presented by Lu et al. with respect to the regulatory approval pathways. Contamination of the seed virus by non-replicative MVA can be easily ruled out by testing the master seed, while Edmonston seed B vaccine strain is the parental virus of the Schwarz strain. The rather limited differences

between the backbones of Edmonston B and Schwarz can be expected to be of considerably less impact than the differences introduced by the exchange of the encoded foreign extra antigen. Therefore, the reviewer feels that this paragraph should be attenuated.

Reviewer #3:

Remarks to the Author:

In this revised manuscript, the authors have dealt satisfactorily with most of the issues raised in my review.

REVIEWERS' COMMENTS

Reviewer #1 (Remarks to the Author):

The Reviewer would like to thank the authors for considering the points made and the extra data provided which significantly enhance the quality and content of this manuscript. Although most points were clearly addressed, some issues may or should still be addressed:

- Considering major point 2 of the original review, the authors indicate in their point-by-point reply the explanation for the loss of antigen-expression in one of their candidates, but the data seemingly have not been incorporated into the manuscript. Since this observation is of general interest for the platform technology, the authors are strongly encouraged to incorporate this information in the final version of their manuscript.

The explanation was added (L.180-188) with additional figures in the supplementary information (S.Fig 4b and S.Fig 4c)

- del Valle et al. should be acknowledged and cited in l. 111 for providing evidence that higher amounts of antigen expressed by recombinant MV triggers better antibody titers

The reference was added in the next paragraph concerning MV expression of the antigen (L.160).

- Suppl. Figs. 6, 9 and 10 are not described in the manuscript's result section. The enclosed information should be transmitted in the text, when necessary, or the Figs should be deleted, if not. The reviewer suggests the first option.

The description for S.Fig 6 and 9 was added (L.235-236 and L.210-212 respectively) while S.Fig 10 was removed and replaced by S.Fig. 11a and b (which is not S.Fig.10).

- Discrepancies between the result text and figure seems to become evident concerning the new hamster experiments depicted in Fig. 8:

a.) panel e: The text states that 7/8 animals had no virus titer in the lungs, but just 5 are

displayed as "not detectable", whereas 2 animals had titers at the limit of detection and 1 revealed low titers. Either the 2 animals at the LOD had some remaining virus (than, this is no absence of virus titer) or they had not - than they should be displayed as "non detectable". Please clarify.

The statement was clarified as suggested. (L.325)

b.) panel f: The text states that 7/8 animals immunized with a single dose showed NABs (l. 296); in the figure, all animals had nAb titers, although one with just a titer of slightly higher than 10. Please clarify.

The statement was clarified as suggested. (L.331)

- Fig. 8f: Why are NAb titers of hamsters with prime-boost, but w/o challenge lower than the titer of hamster cohorts vaccinated once or twice, but later on challenged?

Hamsters' sera were only collected after challenge, the immune response of the challenged groups was also induced by the SARS-CoV2 used in the challenge.

- Fig. 9 / l. 303ff: Please clarify, which hamster cohort exactly is meant by the term "protected animals". Fig. 9 depicts pathologic changes in the "prime only" cohort, at least.

We have reworded (L.346-347) that these hamsters were the prime-boosted animals.

- l. 323f.: "that the 2P mutation enables a stable S expression allowing rMV to replicate better" should be specified. Do the authors mean that a stabilized S is better for MV replication due to absence of fusion activity (as documented in the revised manuscript and elsewhere)?

The statement was specified as suggested. (L.371-372)

- I. 335: The information of the RBD-mutations of the mouse-adapted virus should be included already in the result section.

The details of the RBD mutation was added to the result section. (L.286)

- I. 351: link of broad-spectrum neutralization also of variants to the exposure of hidden epitopes seems quite speculative, here. Also, for e.g. mRNA vaccines, degree of cross-neutralization of variants has been linked to the potency of the induced antibodies against the original virus - better affinity maturation of the antibodies in B cell clones against the pre-fusion conformation could be an exemplary alternative explanation for the observation made.

This alternative explanation was added as suggested. (L.401-402)

- I. 396ff.: The reviewer does not agree on significant obvious disadvantages of the constructs presented by Lu et al. with respect to the regulatory approval pathways. Contamination of the seed virus by non-replicative MVA can be easily ruled out by testing the master seed, while Edmonston seed B vaccine strain is the parental virus of the Schwarz strain. The rather limited differences between the backbones of Edmonston B and Schwarz can be expected to be of considerably less impact than the differences introduced by the exchange of the encoded foreign extra antigen. Therefore, the reviewer feels that this paragraph should be attenuated.

We understand your point. However, according to regulatory aspect, the statement in this paragraph stands true. The GMO qualification and bio-confinement authorization for clinical use of a recombinant MV are granted according to the virus sequence. Our previous clinical experiences proved that using a clinically approved backbone such as the Schwarz strain facilitates a lot. On the manufacturing side, HEp2 cells are not currently accepted by regulatory agencies and MVA elimination strongly impacts the production yield. These steps are difficult to adapt to the industry. Therefore, we think that these points need to be discussed here. Nevertheless, we slightly attenuated the paragraph as suggested. (L.448-451)